# Persistent activity of aerobic methane-oxidizing bacteria in anoxic lake waters due to metabolic versatility

Sina Schorn [1,5] ✉, Jon S. Graf[1], Sten Littmann[1], Philipp F. Hach[1], Gaute Lavik[1], Daan R. Speth [1,2], Carsten J. Schubert [3,4], Marcel M. M. Kuypers [1] & Jana Milucka [1]

Lacustrine methane emissions are strongly mitigated by aerobic methane-oxidizing bacteria (MOB) that are typically most active at the oxic-anoxic interface. Although oxygen is required by the MOB for the first step of methane oxidation, their occurrence in anoxic lake waters has raised the possibility that they are capable of oxidizing methane further anaerobically. Here, we investigate the activity and growth of MOB in Lake Zug, a permanently stratified freshwater lake. The rates of anaerobic methane oxidation in the anoxic hypolimnion reached up to $0.2 \, \mu M \, d^{-1}$. Single-cell nanoSIMS measurements, together with metagenomic and metatranscriptomic analyses, linked the measured rates to MOB of the order Methylococcales. Interestingly, their methane assimilation activity was similar under hypoxic and anoxic conditions. Our data suggest that these MOB use fermentation-based methanotrophy as well as denitrification under anoxic conditions, thus offering an explanation for their widespread presence in anoxic habitats such as stratified water columns. Thus, the methane sink capacity of anoxic basins may have been underestimated by not accounting for the anaerobic MOB activity.

Methane ($CH_4$) is a powerful greenhouse gas, abundantly produced in marine and freshwater environments. Fortunately, its emissions are strongly mitigated by the activity of microorganisms that have the capacity to oxidize methane to $CO_2$[1–3]. These microorganisms, termed methanotrophs, are therefore considered a major biological methane filter. Both archaea and bacteria possess the capacity to activate and oxidize methane, and their relative contribution to methane removal varies in different aquatic habitats[4–6]. In anoxic marine sediments, the majority of methane is consumed by consortia of anaerobic methanotrophic archaea (ANME) and Deltaproteobacteria that couple anaerobic oxidation of methane (AOM) to sulfate reduction[5,7]. Additionally, anaerobic methane oxidation in marine and freshwater habitats can be linked to nitrate or nitrite reduction and is performed either by nitrate/nitrite-dependent ANME archaea[8,9] or by NC10 bacteria related to *Candidatus* Methylomirabilis oxyfera[10–14]. Aerobic methanotrophic activity is typically attributed to methane-oxidizing bacteria (MOB) belonging to the Alpha- or Gammaproteobacteria[15]. These bacterial methanotrophs activate methane via a soluble or particulate methane monooxygenase that catalyzes the initial oxidation of methane to methanol using molecular oxygen. Methane oxidation by NC10 bacteria is also mediated by a methane monooxygenase and, therefore, oxygen-dependent. However, these bacteria couple methane oxidation to denitrification via a unique nitrite-dependent methane oxidation pathway that produces oxygen intracellularly via nitric oxide dismutation[11], which enables NC10 bacteria to thrive in anoxic environments[12–14,16,17].

[1]Max Planck Institute for Marine Microbiology, Bremen, Germany. [2]Division of Microbial Ecology, Center for Microbiology and Environmental Systems Science, University of Vienna, Vienna, Austria. [3]Department of Surface Waters, Swiss Federal Institute of Aquatic Science and Technology (Eawag), Kastanienbaum, Switzerland. [4]Institute of Biogeochemistry and Pollutant Dynamics, ETH Zurich, Zurich, Switzerland. [5]Present address: Department of Marine Sciences, University of Gothenburg, Gothenburg, Sweden. ✉e-mail: sina.schorn@gu.se

With the exception of NC10 bacteria, bacterial methane oxidation is considered an obligate oxygen-dependent process constrained to oxic environments. However, it has been shown recently that various taxa of MOB differ in their oxygen requirements and inhabit distinct niches in the water column[18,19]. Gammaproteobacterial MOB (gamma-MOB) typically exhibit the highest activity under hypoxic conditions[20,21] and, therefore, thrive primarily at the oxic–anoxic interface in the water column or sediment, where they experience high fluxes of both methane and oxygen[22].

Interestingly, gamma-MOB belonging to the ubiquitous gammaproteobacterial order Methylococcales are also commonly detected in anoxic waters or sediments of stratified lakes and marine basins[18,23–32]. The abundance of gamma-MOB in anoxic waters is poorly understood because there is so far no mechanistic explanation for methane oxidation by these microorganisms in the complete absence of oxygen. It has therefore been proposed that even in anoxic environments, the activity of MOB might be linked to the periodic occurrence of trace amounts of oxygen (e.g. refs. 24,33), which are undetectable with current techniques[34]. For example, in shallow lakes, where light penetrates into anoxic water depths, aerobic methane oxidation can be sustained through in situ oxygen production by photosynthetic algae[27,35]. Lateral and/or vertical intrusions of oxic waters into anoxic depths were also proposed to sustain methanotrophic activity in situ[24]. Additionally, some gamma-MOB evolved specific mechanisms to minimize the overall oxygen demand of the cell by either effectively directing oxygen towards particulate methane monooxygenase (pMMO) activity[36–39] or by performing anaerobic respiration[31,33,40–42]. Denitrification, the stepwise reduction of nitrate or nitrite to $N_2$ or $N_2O$, has been commonly invoked as a mechanism allowing for the survival of bacterial MOB under oxygen limitation. This metabolism has so far only been demonstrated experimentally for *Methylomonas denitrificans* cultures[41], but many cultured and environmental gamma-MOB possess the genes for respiratory nitrate reduction[19,33,42,43]. Alternatively, cultured methanotrophs of the genus *Methylomicrobium* have been shown to conserve energy through fermentation-based methanotrophy rather than aerobic respiration or denitrification, thus

representing another strategy to overcome oxygen limitation[38]. In this process, MOB switch to a novel fermentation mode at low oxygen tensions and convert methane to fatty acids (including formate, acetate, succinate, lactate, and hydroxybutyrate) and hydrogen, via primary oxidation of methane to formaldehyde and pyruvate, followed by fermentation of these compounds. This process would have important implications for lacustrine methane cycling in that it would allow methane carbon to be retained in anoxic waters in the form of microbial biomass[44]. Recent studies suggested a broad prevalence of this metabolism among gammaproteobacterial MOB in lake water columns based on metagenomic analyses[45,46]. Irrespective of the utilized energy-conserving pathway (anaerobic respiration or fermentation), in all cases, molecular oxygen is still thought to be required for the initial activation of methane via the pMMO.

A widespread capacity of gamma-MOB for thriving anaerobically in anoxic environments would bear great significance for the in situ activity of this important microbial group. Yet, at this point, the occurrence and prevalence of gammaproteobacterial 'anaerobic' methane oxidation in the environment remains elusive. Here, we investigated the activity and growth of different aerobic gammaproteobacterial and *Methylomirabilis*-like methanotrophs in the water column of Lake Zug, a deep, permanently stratified freshwater lake in central Switzerland, using a combination of stable isotope incubations, single-cell imaging mass spectrometry, as well as metagenomic and metatranscriptomic analyses.

## Results
### Water column chemistry and methane fluxes in the hypolimnion of Lake Zug
In September 2017, the oxic–anoxic interface in the water column of Lake Zug was located at a depth of ca. 121 m. Below this depth, oxygen concentrations decreased below the detection limit of 20 nM down to the lowest sampling depth of 180 m (Fig. 1a). Nitrate was detected throughout most of the anoxic hypolimnion and concentrations generally decreased with depth, from 21 µM at 120 m (oxic–anoxic interface) to 0.2 µM at 180 m. Nitrite was also detected in the water column,

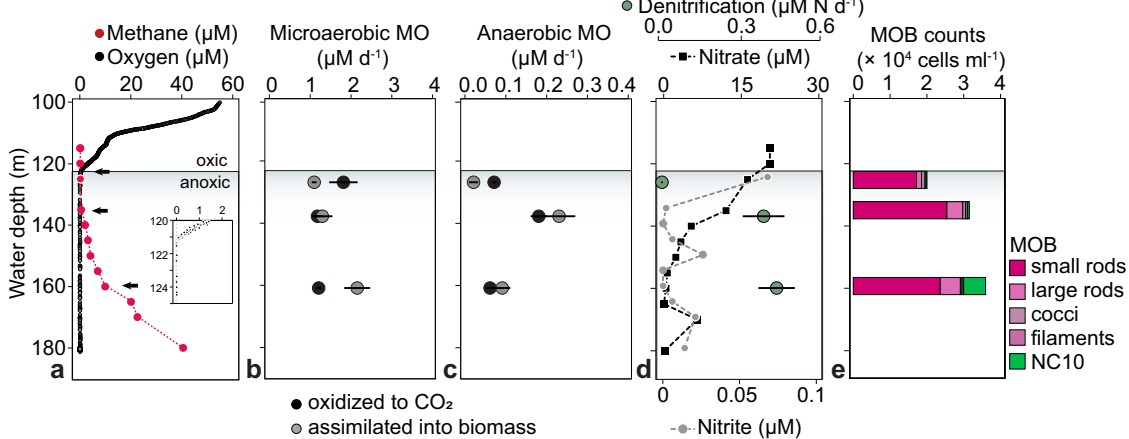

**Fig. 1 | Lake Zug water column profiles of dissolved gasses, inorganic nutrients, methane oxidation rates, denitrification rates, and cell counts of methane-oxidizing bacteria (MOB). a** In situ concentration profiles of oxygen (black) and methane (red) at water depths below 100 m. Black arrows indicate sampling depths for stable isotope incubations and single-cell analyses. Inlet shows oxygen concentrations at the oxic–anoxic interface. **b** Rates of microaerobic methane oxidation (MO) to carbon dioxide (black symbols) and to biomass (gray symbols). **c** Rates of anaerobic methane oxidation to carbon dioxide and methane carbon assimilation to biomass. Note that rates of anaerobic methane oxidation are approximately 10-fold lower than those of aerobic methane oxidation. The rate of methane carbon assimilation at 123 m depths was statistically not significant

($p = 0.05$) due to a non-linear increase of $^{13}C$ over time. **d** In situ concentrations of nitrate (black symbols) and nitrite (gray symbols) and denitrification rates (green symbols). Statistically significant methane oxidation, assimilation, and denitrification rates in panels (**b**–**d**) (one-sided $t$-test, $p < 0.05$) are presented as mean values ± SEM calculated from the linear regression of the first five time points across 8 days of incubation. **e** MOB cell counts in the anoxic hypolimnion of Lake Zug based on CARD FISH, comprising gamma-MOB (stained with probes Mγ84 and Mγ705) and NC10 bacteria (stained with probe DBACT-1027). The gamma-MOB population comprised morphologically distinct cell types, which were classified as small rods, large rods, cocci, and filaments. Source data are provided as a Source Data file.

albeit at concentrations below 30 nM in most depths, with the exception of the uppermost sampling depth located directly below the oxycline, where nitrite concentrations were about 70 nM (Fig. 1d).

Methane concentrations were highest in the anoxic hypolimnion (ca. 40 μM at 180 m) and decreased gradually towards the oxic–anoxic interface with less than 1 μM methane detected ca. 15 m below the interface (Fig. 1a). The methane profile showed gradual decrease in concentrations with three distinct zones that were located between the deepest sampling depth close to the sediment surface at 180 and 160 m, between 160 and 135 m, and between 135 and 120 m. Assuming steady-state conditions, the shape of the concentration profile suggested three putative zones of methane consumption, one at the oxic–anoxic interface and two below. We calculated methane fluxes for these three zones and found that the highest methane flux of 3.6 mmol $CH_4$ $m^{-2}$ $d^{-1}$ occurred between the sediment surface and 160 m. Between 160 and 135 m, the methane flux was lower with 0.9 mmol $CH_4$ $m^{-2}$ $d^{-1}$ and lowest, with 0.05 mmol $CH_4$ $m^{-2}$ $d^{-1}$, around the base of the oxycline between 135 and 120 m.

## Bulk methane oxidation and denitrification rates

Bulk methane oxidation and denitrification rates were determined in three water depths, located at 123 m (oxic-anoxic interface), 135 m, and 160 m (deep, anoxic and methane-rich waters). From all three depths, hypoxic (ca. 10 μM $O_2$ added) and anoxic (degassed and no $O_2$ added, 20 μM $^{15}$N-nitrate) $^{13}$C-methane-amended incubations were set up, from which the rates of methane oxidation to $CO_2$ and nitrate reduction to $N_2$ (denitrification) were determined.

At all three depths, methane oxidation rates in the hypoxic incubations were higher than in the anoxic ones (Fig. 1b, c). In hypoxic incubations, the highest methane oxidation activity was measured in the uppermost depth at 123 m (1.81 μM $d^{-1}$), but the $^{13}CO_2$ production showed an initial 2-day lag phase and only increased linearly afterward, suggesting that the methanotrophic community needed time to adapt to the incubation conditions, most likely because of the low in situ methane concentrations and fluxes at the time of sampling (Fig. S3). If the lag phase was omitted from the rate calculation, the methane oxidation rate increased slightly to 2.09 μM $d^{-1}$. Hypoxic methane oxidation rates in both anoxic depths were approximately twofold lower, with 1.17 μM $d^{-1}$ at 135 m and 1.20 μM $d^{-1}$ at 160 m (Fig. 1b). However, at these depths, $^{13}CO_2$ production started immediately and proceeded linearly throughout the course of the incubation (Fig. S3).

In anoxic, nitrate-supplemented incubations, methane oxidation rates were highest at 135 m with 0.18 μM $d^{-1}$ and lower at the depths above and below (0.07 μM $d^{-1}$ at 123 m and 0.06 μM $d^{-1}$ at 160 m) (Fig. 1c). At all three depths, methane was oxidized immediately and the rates proceeded linearly over the 8 days of incubation. Denitrification rates, measured as $^{30}N_2$ production from $^{15}$N-labeled nitrate, increased with depth, from 0.02 μM N $d^{-1}$ at 123 m, to 0.41 μM N $d^{-1}$ at 135 m, and 0.46 μM N $d^{-1}$ at 160 m (Fig. 1d), with no apparent correlation between the measured denitrification and methane oxidation rates (see also section "Discussion").

## Bulk methane carbon assimilation rates

In addition to the bulk rates of methane oxidation to $CO_2$, we also determined the amount of methane-derived carbon that was assimilated into total microbial biomass from the same incubations (Fig. 1b, c). Surprisingly, we found that methane carbon assimilation into biomass equaled or surpassed methane oxidation to $CO_2$ in all incubations with water collected from both anoxic depths (Table S1). More specifically, under hypoxic conditions we measured methane carbon assimilation rates of 1.29 μM $d^{-1}$ at 135 m and 2.15 μM $d^{-1}$ at 160 m, corresponding to 52% and 64% of the methane carbon being incorporated into microbial biomass at the respective depths as compared to $CO_2$ production. Similar values were obtained from anoxic incubations, where assimilation rates of 0.23 μM $d^{-1}$ at 135 m

and 0.09 μM $d^{-1}$ at 160 m were determined, corresponding to 56% and 60% of methane consumed being recovered as biomass, respectively. For comparison, at the oxic–anoxic interface, 38% (hypoxic) and 22% (anoxic) of methane carbon were recovered as biomass. High methane assimilation rates in the anoxic hypolimnion (albeit not exceeding methane oxidation) were also measured for two consecutive years (Fig. S1; Supplementary Note 1).

## Abundance of MOB across different oxygen regimes

Gammaproteobacterial methane-oxidizing bacteria (gamma-MOB) of the order Methylococcales were abundantly present in the water column of Lake Zug in 2017, as inferred from CARD FISH-based cell counts at regular depth intervals between 115 and 180-m depths (Fig. S2). In general, the abundance of gamma-MOB slightly increased with depth, from ca. $2.0 \times 10^4$ cells $ml^{-1}$ at 125 m to $3.0 \times 10^4$ cells $ml^{-1}$ at 160 m (Fig. 1e). Based on cell morphology we defined three distinct groups of unicellular gamma-MOB: coccoid cells (avg. cell volume ca. 3.2 μm³), small rod-shaped cells (ca. 1.6 μm³) and large rod-shaped cells (ca. 7.4 μm³). Additionally, long Crenothrix-like filaments (avg. filament volume ca. 55.3 μm³) were observed in the water samples (Fig. 2a). Crenothrix-like filaments were not targeted by the used CARD FISH probes but due to their characteristic morphology could be identified in microscopic analyses, thus comprising a fourth group of gamma-MOB in the incubations. Based on cell counts, the small rod-shaped gamma-MOB were highest in abundance and showed an even distribution throughout all analyzed water depths. Also, clusters of coccoid cells were distributed across the anoxic water column without a clear distribution pattern (Fig. S2). In contrast, large rod-shaped cells showed the highest abundance between 135 and 160 m, and Crenothrix-like filaments increased in abundance with depths. We also detected NC10 bacteria in the hypolimnion of Lake Zug by CARD FISH using probe DBACT-1027, which visualized 0.9 μm long rods with average cell volumes of 0.14 μm³ ($n = 20$), which also increased in abundance from ca. $3.3 \times 10^2$ cells $ml^{-1}$ at 125 m to $6.0 \times 10^3$ cells $ml^{-1}$ at 160 m (Fig. 1e).

## Cell growth and methane assimilation under hypoxic and anoxic conditions

Cell numbers of all four gamma-MOB groups increased in hypoxic incubations, whereas in anoxic incubations only rod-shaped MOB cell numbers increased (from $1.4 \times 10^4$ cells $ml^{-1}$ at the start of the experiment to $3.7 \times 10^4$ cells $ml^{-1}$ after 8 days of incubation). To confirm that the observed growth of gamma-MOB in our incubations was methane-derived, we quantified the assimilation of $^{13}$C-labeled methane carbon into the biomass of the different MOB groups using nanoscale secondary-ion mass spectrometry (nanoSIMS). Under hypoxic conditions, Crenothrix-like filaments showed the highest $^{13}$C assimilation per cell of, on average, 29 at.% $^{13}$C excess within the first 24 h of incubation, corresponding to a $^{13}$C-based growth rate of 0.49 $d^{-1}$ (assuming that they only grew on methane as a carbon source). Other gamma-MOB groups, i.e. large rods, small rods, and cocci, showed lower, albeit still high $^{13}$C assimilation of 24, 21, 23 at.% $^{13}$C after 24 h, corresponding to growth rates of 0.39, 0.34, 0.38 $d^{-1}$, respectively (Fig. 2b, Table S2).

When the calculated excess $^{13}$C at.% values were converted to cellular C assimilation rates, more profound differences in single-cell methane assimilation rates could be observed for the four gamma-MOB groups. Under hypoxic conditions, the highest cellular C assimilation rates were exhibited by the Crenothrix-like filaments (30.1 fmol $^{13}$C $cell^{-1}$ $d^{-1}$), followed by large rods (9.8 fmol $^{13}$C $cell^{-1}$ $d^{-1}$), cocci (6.4 fmol $^{13}$C $cell^{-1}$ $d^{-1}$), and small rods (4.2 fmol $^{13}$C $cell^{-1}$ $d^{-1}$).

In anoxic incubations, only the large rod-shaped MOB exhibited a high rate of $^{13}$C-methane assimilation, which was identical to the rates in the hypoxic incubations (ca. 21 at.% $^{13}$C per cell within 24 h or 8.6 fmol $^{13}$C $cell^{-1}$ $d^{-1}$). Importantly, the $^{13}$C content in the large rods increased evenly throughout the incubation period, under both

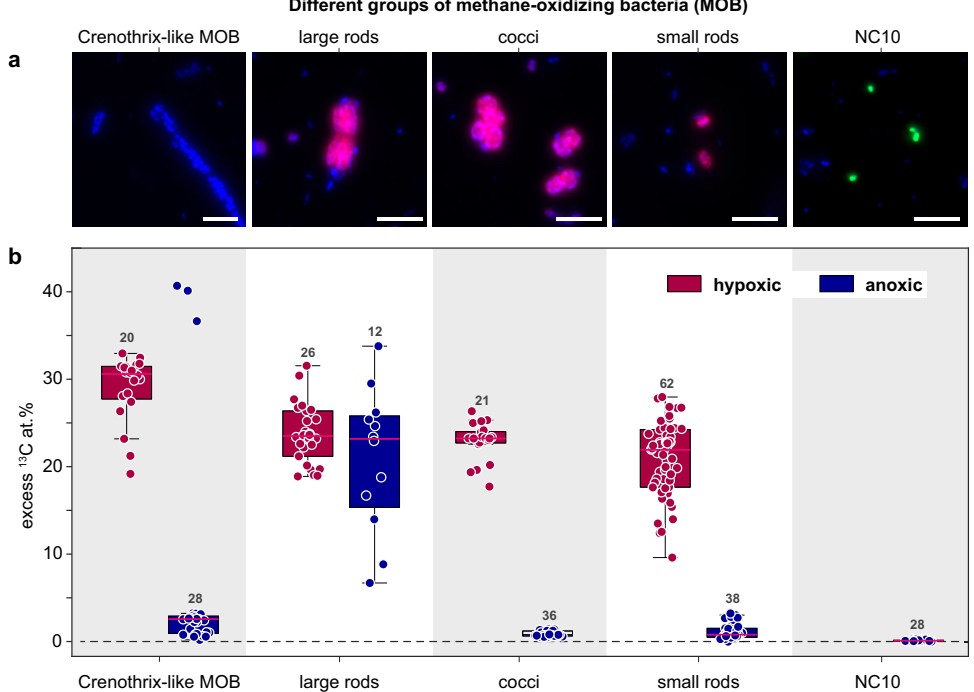

**Fig. 2 | Morphology and activity of methane-oxidizing bacteria under hypoxic and anoxic conditions. a** CARD FISH images of gammaproteobacterial MOB stained with probes Mγ84 and Mγ705 (red) and NC10 bacteria stained with probe DBACT−1027 (green). All cells were counterstained with 4′,6-diamidino-2-pheny-lindole (DAPI; blue). *Crenothrix*-like filaments were identified based on their mor-phology after DAPI staining. Gamma-MOB were classified based on their size and shape as large rods, cocci, and small rods (from left to right). **b** Excess $^{13}$C at.% enrichment of MOB after incubation with $^{13}$C-labeled methane for 24 h under hypoxic (red) and anoxic (blue) conditions determined by nanoSIMS analysis. $^{13}$C at.% values are given as excess values for each analyzed cell. The 0 value on the *y*-axis depicts the natural abundance of $^{13}$C (1.1%). The number of cells analyzed per category (n) is shown as scatter and indicated above each boxplot. Boxplots depict the 25–75% quantile range, with the center line representing the median (50% quantile) and whiskers representing the 5 and 95 percentile. All analyses were performed on samples from 135 m water depth. Scale bar represents 3 μm. Source data are provided as a Source Data file.

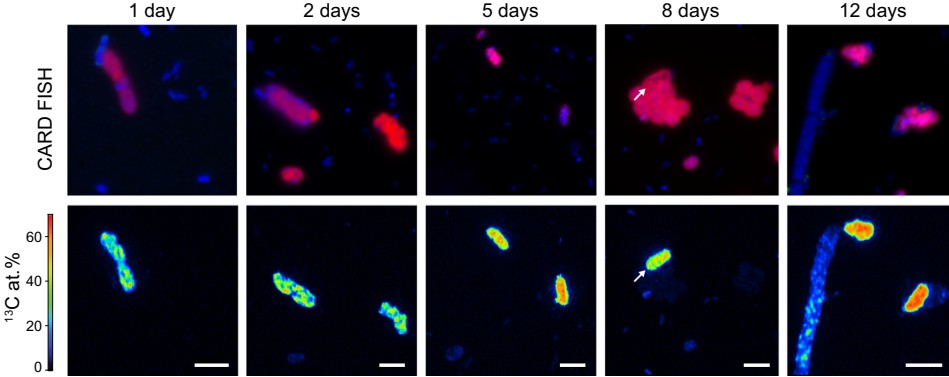

**Fig. 3 | Cellular $^{13}$C enrichment of large rod-shaped MOB in anoxic incubations over time.** CARD FISH images (using probes Mγ84 and Mγ705) and corresponding nanoSIMS measurements of gamma-MOB showing substantial $^{13}$C enrichment under anoxic conditions over 12 days of incubation. Scale bar represents 3 μm.

hypoxic and anoxic conditions (Fig. 3). Combined, this suggests that the bulk of the methane oxidation under apparent anoxia was due to the activity of the large rod-shaped gamma-MOB. Moreover, as the $^{13}$C assimilation per cell was essentially identical under the two conditions, the analyzed rod-shaped cells appeared to grow at comparable rates under hypoxic and anoxic conditions (Fig. 4). Apart from the large rods, *Crenothrix*-like filaments also showed some $^{13}$C assimilation in anoxic incubations; however, with a much lower rate (median enrich-ment of 2.6 at.% $^{13}$C per cell) than the large rods. In contrast to the large rods, the degree of $^{13}$C enrichment strongly varied among individual *Crenothrix*-like filaments and cells (Fig. S9; Supplementary Note 3). The coccoid and small rod-shaped MOB showed even lower $^{13}$C-methane

assimilation of about 1 at.% $^{13}$C per cell after 24 h of anoxic incubation, which was considerably lower than the assimilation observed for the large gamma-MOB rods. NC10 bacteria exhibited the lowest $^{13}$C enrichment among all MOB, with 0.1 at.% $^{13}$C per cell after 24 h, which marginally increased to 0.3 at.% $^{13}$C per cell after 12 days of incuba-tion (Fig. S7).

As anoxic incubations were also supplemented with $^{15}$N-labeled nitrate for the quantification of denitrification rates, we measured the assimilation of $^{15}$N into the gamma-MOB cells as another proxy for their activity. We detected the highest uptake of $^{15}$N by the large rod-shaped cells, consistent with the $^{13}$C data (Fig. S6). Substantially lower, but still detectable, $^{15}$N enrichments were observed for the other gamma-MOB

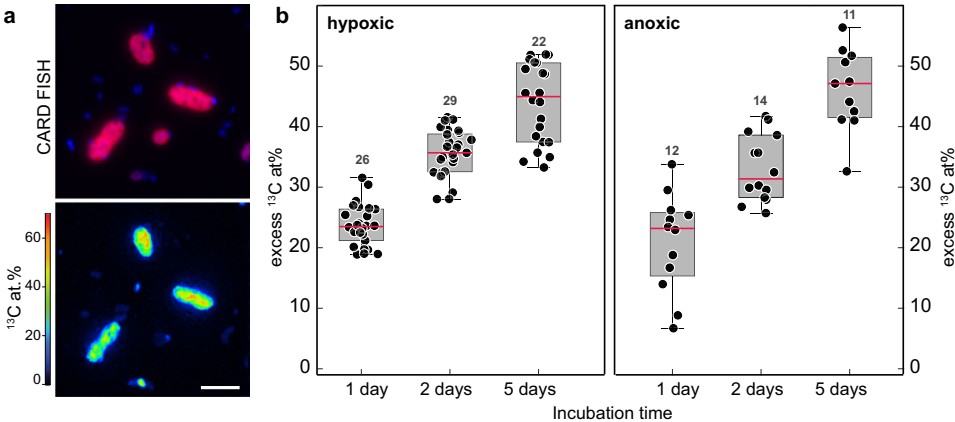

**Fig. 4 | Activity of large rod-shaped gamma-MOB under anoxic conditions determined by nanoSIMS analyses. a** CARD FISH image (using probes Mγ84 and Mγ705) and corresponding nanoSIMS measurements of large rod-shaped gamma-MOB (red) showing substantial $^{13}C$ enrichment under anoxic conditions. Images and measurements as shown in (**a**) were obtained from samples collected after 8 days of anoxic incubation. **b** Excess $^{13}C$ at.% of large rod-shaped gamma-MOB after incubation with $^{13}C$-labeled methane for 1, 2, and 5 days under hypoxic (left) and anoxic (right) conditions. $^{13}C$ at.% values are given as excess values for each analyzed cell. The 0 value on the *y*-axis depicts the natural abundance of $^{13}C$ (1.1%). The number of cells analyzed per category (*n*) is shown as scatter and indicated above each boxplot. Boxplots depict the 25–75% quantile range, with the center line representing the median (50% quantile, highlighted in red) and whiskers representing the 5 and 95 percentile. Source data are provided as a Source Data file.

groups indicating that they were active too, albeit to a lesser extent. However, the varying preferences for different N sources among the different MOB groups, such as for ammonium that is abundantly present in the lake water, could also contribute to the observed differences.

## Metagenomic and metatranscriptomic analyses

Metagenomic analyses of the microbial community composition were performed for the three incubation depths. Based on the classification of small subunit (SSU) ribosomal RNA gene reads obtained from the metagenomes we detected members of the Methylococcales in all three investigated water depths, comprising mainly the families Methylomonadaceae and, to a lesser extent, Methylococcaceae. The relative abundance of 16S rRNA gene sequences assigned to the Methylococcales within the microbial community (considering all bacterial and archaeal sequences) markedly increased from ca. 1% at 123 m, to ca. 7% at 135 m, and ca. 20% at 160 m (Fig. 5a), although it should be noted that these estimates could be biased due to e.g. multiple 16S gene copy numbers possessed by MOB. We also detected anaerobic bacterial methanotrophs of the phylum NC10 belonging to the genus *Candidatus* Methylomirabilis in our metagenomic datasets with relative abundances of 0.04% (123 m), 1.3% (135 m), and 1.4% (160 m) based on the classification of small subunit (SSU) ribosomal RNA gene reads. Anaerobic methanotrophic archaea were mostly absent with only a few reads detected (Supplementary Dataset 4).

From our three metagenomic datasets we assembled 14 bins that contained a particulate methane monooxygenase gene (*pmoABC*), the key marker gene for bacterial methanotrophs. One of these bins closely affiliated with *Candidatus* Methylomirabilis limnetica of the NC10 phylum, the other 13 bins affiliated with the Methylococcales gamma-MOB, more specifically with Methylobacter A and Methylovulum, as well as with uncultured clades SXIZ01, KS41, UBA4132 and UBA10906 (Figs. 5 and S5; Supplementary Note 2). None of the bins contained a 16S rRNA gene, however, we recovered ten 16S rRNA genes belonging to gamma-MOB of the Methylococcales order from the metagenomes (2 sequences from 123 m, 3 from 135 m, and 5 from 160 m), which showed a similar taxonomic affiliation as the bins (Supplementary Note 2, Supplementary Dataset 1).

In order to identify the methanotrophs that are responsible for the observed activity, we screened the *pmoABC*-containing bins for genes coding for anaerobic metabolism, i.e. denitrification and fermentation (Supplementary Dataset 5). Overall, most bins contained a subset of denitrification genes, including nitrate reductases (*narGHIJ* and *napAB*), nitrite reductases (*nirKS*), and nitric oxide reductase (*norBC*), indicating the potential for partial denitrification of this gamma-MOB. Notably, the absence of the nitrous oxide reductase (*nosZ*) gene in all bins suggests that the Lake Zug MOB lack the ability to further reduce $N_2O$ to $N_2$ (Fig. 5b). In the NC10 bin only two genes related to denitrification (i.e. *nirS* and qNOR) were detected.

Additionally, the NC10 bin as well as 12 out of the 13 Methylococcales bins, contained several genes involved in fermentation. More specifically, we detected all genes of the mixed acid fermentation pathway, which have been proposed previously to allow MOB to metabolize methane-derived compounds via fermentation[38]. This included genes encoding for lactate dehydrogenase (*ldhA*), phosphate acetyltransferase (*pta*), acetate kinase (*ackA*), malic enzyme (*sfcA*), malate dehydrogenase (*mdh*), fumarate hydratase (*fumC*), and succinate dehydrogenase (*sdhABCD*).

We next investigated the transcription of fermentation and denitrification genes in metatranscriptomes collected from the three incubation depths. Overall, all bins, except bin 7, showed transcription of fermentation and denitrification genes. Interestingly, transcripts of fermentation genes were almost exclusively retrieved from the deepest investigated water depth (160 m), at which anoxic, methane-rich conditions prevailed (Fig. 5b). Transcripts of denitrification and fermentation genes were also detected in metatranscriptomes from the shallower depths, albeit only sporadically (Supplementary Dataset 5). Importantly, high transcription of *pmoABC* genes was detected at all investigated water depths for most of the 14 bins.

## Discussion

In stratified lacustrine and marine water columns, aerobic methane oxidation at the oxic–anoxic interface is typically a major sink for upwards-diffusing methane and, thus, an important biological methane filter[18,28,30,32,47]. During our sampling campaign in September 2017, hypoxic incubations from the oxic–anoxic interface (123 m depth) displayed high potential rates of methane oxidation (up to 1.8 μM d$^{-1}$). However, the rates were preceded by a ca. 2-day long lag phase, indicating a necessity for an adaptation period for the methanotrophic community, potentially due to low in situ activity at the time of sampling. Indeed, the in situ methane concentration profiles showed that very little methane diffused all the way up to the

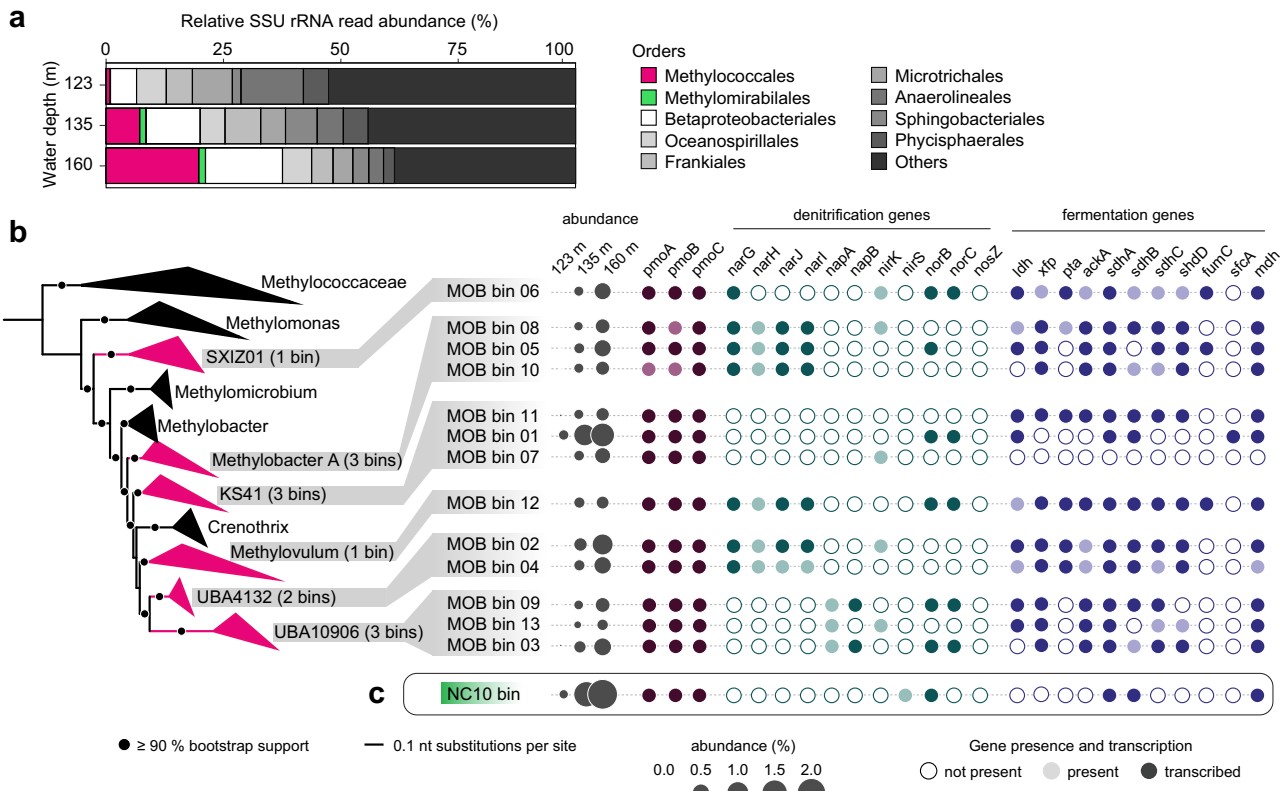

**Fig. 5 | Abundance, phylogeny, and metabolic capacity of methane-oxidizing bacteria in the hypolimnion of Lake Zug. a** Relative small subunit (SSU) rRNA read abundance of the microbial community from the three incubation depths classified at order level. Methylococcales-related sequences are highlighted in red, sequences belonging to the order Methylomirabilales of the NC10 phylum are highlighted in green. **b** Methylococcales genome tree showing genera to which *pmoA*-containing bins recovered from the metagenomes affiliate. The branches are colored if the genus contains at least one of the bins (number of bins indicated in brackets). Circles on branches represent 90% and more bootstrap support. Abundance of bins (in percent) is inferred from read mapping. Metagenomic and metatranscriptomic analysis is shown for samples collected from the anoxic, methane-rich hypolimnion (160 m). The transcription of genes is depicted as filled circles, representing genes associated with methane oxidation, denitrification, and mixed acid fermentation pathways in the respective bins. Circles indicate absence (white) or presence (gray) of genes in the bins, as well as gene transcription (colored). **c** Abundance of the NC10 bin as well as presence and transcription of genes related to *pmoABC*, denitrification, and fermentation. Particulate methane monooxygenase (*pmoABC*), membrane-bound nitrate reductase (*narGHIJ*), periplasmic nitrate reductase (*napAB*), nitrite reductase (*nirKS*), nitric oxide reductase (*norBC*), nitrous oxide reductase (*nosZ*), lactate dehydrogenase (*ldh*), phosphate acetyltransferase (*pta*), acetate kinase (*ackA*), succinate dehydrogenase (*sdhABCD*), fumarate hydratase (*fumC*), NAD-dependent malic enzyme (*sfcA*), malate dehydrogenase (*mdh*). Source data are provided as a Source Data file.

oxic–anoxic interface; instead, most methane was consumed already 10–15 m below, suggesting a likely sink for methane in the anoxic hypolimnion at and below 135 m water depth (Fig. 1a). Correspondingly, high and linear rates of anaerobic methane oxidation (up to 0.2 μM d⁻¹) with immediate onset could be detected in the anoxic incubations from both deep water depths (135 and 160 m; Fig. 1c). For comparison, the anaerobic rates were 5 to 20-fold lower than rates from parallel hypoxic incubations with water from these same water depths, to which low concentrations of oxygen (ca. 10 μM O₂) were added (Fig. 1b).

Despite the evidence for ongoing, apparently anaerobic, methane oxidation, anaerobic methane-oxidizing archaea were not detected in the deep anoxic water depths. Instead, the methane-oxidizing community in the anoxic waters was dominated by aerobic gammaproteobacterial methanotrophs (gamma-MOB) (Fig. 1e) as well as denitrifying methanotrophs of the NC10 phylum. This is in agreement with previous analyses of the methanotrophic community in this lake[12,28]. Morphologically, four distinct cell types could be distinguished among the gamma-MOB targeted with a specific FISH probe (Fig. 2a), of these *Crenothrix*-like filaments[42] and small cocci (presumably *Methylovulum*-like MOB) were numerically least abundant and their cell numbers increased (*Crenothrix*) or remained constant (small cocci) throughout the water column (Figs. 1e and S2;

Supplementary Note 2). Most abundant were small rods, whose abundance decreased with increasing depth. In contrast, large rods— which, due to their large size, constituted a large proportion of the MOB biomass (Fig. S2)—were more abundant below the oxic–anoxic interface (Fig. 1e). The conspicuous morphology of the large rods detected by a gamma-MOB-specific FISH probe is strongly reminiscent of methanotrophs from the *Methylobacter* genus of the Methylococcales order. Cultured representatives of *Methylobacter tundripaludum* are typically large, rod-shaped and free-living[48,49], although other morphologies have been observed for other species[46]. Environmental *Methylobacter*-related methanotrophs are frequently detected in hypoxic and anoxic freshwater environments[24,26,31,50]. Indeed, our molecular analyses confirmed the presence and abundance of *Methylobacter* spp. in all three incubation depths (Fig. 5b). Multiple genomic bins assigned or related to *Methylobacter* spp. were retrieved from our samples, and their relative abundance increased with depth, supporting the prevailing opinion that members of this genus thrive in low-oxygen environments and might play a key role in methane removal there[29,31].

In our anoxic incubations that were setup to mimic the conditions in the anoxic hypolimnion (no added O₂, added nitrate, excess methane), we consistently detected growth and activity of the *Methylobacter*-like MOB; in fact, these were the only MOB exhibiting

substantial growth on methane in the absence of oxygen (Fig. 3). Surprisingly, the activity of these MOB—with regards to cell-specific $^{13}C$ methane assimilation rates (9.8 vs. 8.6 fmol $^{13}C$ cell$^{-1}$ d$^{-1}$) or $^{13}C$-based growth rate (0.39 vs. 0.34 d$^{-1}$)—did not significantly differ with or without oxygen (Table S2), suggesting that their activity is not suppressed under oxygen-limiting conditions. This is in striking contrast to the other three MOB morphotypes, which exhibited comparable rates of growth and activity only when oxygen was added to the incubations. The putative *Crenothrix*-like filaments occasionally also exhibited high $^{13}C$ enrichment under anoxic conditions, although less consistently than the large Methylobacter-like MOB. However, *Crenothrix*-like MOBs were solely identified based on their filamentous cell shapes; it is therefore possible that not all of the measured filaments were indeed *Crenothrix* (Supplementary Note 3). These results suggest that while high diversity exists among methane-oxidizing bacteria under hypoxic conditions, distinct *Methylobacter*-like MOB are largely responsible for methane oxidation in the anoxic waters of Lake Zug. Importantly, this implies that the lower rates of anaerobic methane oxidation measured in the hypolimnion compared to the oxic–anoxic interface were a consequence of a lower number of MOB that remained active under anoxic conditions rather than a general decrease in cell-specific methanotrophic activity of MOB in the anoxic hypolimnion.

It should be noted that using $^{13}C$-methane assimilation as a proxy for activity precludes us from evaluating the potential contribution of *Ca*. Methylomirabilis-like bacteria in our samples. This is because these methanotrophs preferentially grow on inorganic carbon[51] and thus did not get enriched in $^{13}C$ in our incubations (Fig. 2b). However, based on their uptake of $^{15}N$-nitrate over time (Fig. S7) and their moderate abundance in the deep water (Fig. 1e) we assume that they may also contribute to methane oxidation and denitrification in the Lake Zug hypolimnion, even under non-bloom conditions, similar to other lakes[14].

It remains to be elucidated why the *Methylobacter*-like MOBs are so notably more successful in anoxic environments than other MOB, as well as how their activity can be sustained in the absence of oxygen. Importantly, we do not rule out the presence of trace amounts of oxygen in our ex-situ bottle incubations, either originating from microbial production or oxygen contamination during sample handling (Supplementary Note 5). In fact, based on our current understanding of methane oxidation by bacteria, this trace oxygen is likely indispensable for the initial oxidation of methane to methanol by methane monooxygenase. In this regard it is interesting to note that also aerobic ammonium oxidizers, which use a related enzyme, ammonia monooxygenase, appear capable of utilizing traces of oxygen to sustain their activity in oxygen-deficient environments[52].

In the environment, oxygen is most likely introduced at low concentrations periodically through e.g. intrusions and mixing, which is to some extent a possible scenario also for Lake Zug[53]. Alternative biological mechanisms have been proposed, such as dark oxygen production by NC10 bacteria[11,54] and ammonium-oxidizing archaea (AOA)[55], as well as oxygen production via water splitting[56]. However, the relevance of these processes as a potential oxygen source in anoxic environments remains to be determined.

Alternatively, some methanotrophs appear to be capable of fermentation-based methanotrophy under low oxygen conditions, a process, in which methane carbon is converted into fatty acids, with little biomass synthesis[38]. We indeed found that all MOB bins detected in our lake metagenomes, except bin 7, encoded as well as transcribed several fermentation genes, including those of the mixed acid fermentation pathway (Fig. 5b). This is in agreement with the observations that many bacterial methanotrophs possess the genetic repertoire for fermentation, including lake taxa[19]. Our transcriptome data support the notion that fermentation-based methanotrophy may indeed be employed in situ, preferentially in anoxic waters. The

employment of the fermentation-based methanotrophy would minimize the oxygen requirements of the MOB while simultaneously leading to methane carbon being retained in the low-oxygen waters in the form of microbial biomass instead of being oxidized to $CO_2$.

Indeed, our analyses show that methane carbon was efficiently assimilated into microbial biomass, likely via direct assimilation of methane by proteobacterial methanotrophs as well as uptake of excreted fermentation products by non-methanotrophs. Interestingly, in contrast to the proteobacterial MOB, we did not detect substantial uptake of $^{13}CH_4$-derived carbon by the anaerobic *Ca*. Methylomirabilis methanotrophs (Fig. 2b), in agreement with their proposed growth on inorganic carbon[51]. These data show that despite their high abundance and proposed activity in anoxic waters, *Ca*. Methylomirabilis-like bacteria do not contribute to the retention of methane carbon in the lake hypolimnion by direct assimilation.

Methane carbon retention seemed to be particularly pronounced in the anoxic depths, where rates of methane carbon assimilation matched or even exceeded methane oxidation to $CO_2$ (Fig. 1b, c; Table S1). These data strongly suggest that 'fermentation-based methanotrophy' occurred under the investigated conditions, thus enhancing the retention of methane carbon in the hypolimnion. Additionally, the production of volatile fatty acids may, in turn, support the growth and activity of mixo- and heterotrophic microbial communities. It should be noted that the partitioning of methane-derived carbon between biomass assimilation and complete oxidation to carbon dioxide may depend on the environmental conditions as well as the composition of the methanotrophic community. Indeed, even in Lake Zug, the average proportion of assimilated methane carbon varied between years, with methane carbon assimilation contributing less to total methane turnover in October 2018 and May 2019 than in September 2017. In any case, our data provide first indications for the environmental implications of the proposed fermentation process that bears great relevance to our understanding of the cycling of methane-derived carbon in anoxic environments. Additionally, our results suggest that a high proportion of methane carbon can be assimilated in addition to being oxidized to carbon dioxide under hypoxic and anoxic conditions (Fig. 6). Therefore, the methane sink capacity of anoxic basins may be underestimated, with up to 60% of methane carbon being retained in the anoxic hypolimnion in the form of biomass.

In addition to fermentation, some aerobic MOBs have been proposed to be capable of conserving energy through nitrate or nitrite respiration[41]. Most cultured gamma-MOB encode some genes for respiratory nitrate/nitrite reduction in their genomes[43], which enable them to switch to anaerobic respiration when oxygen becomes limiting[42,57,58]. Environmental data also support a link between methane oxidation and denitrification, e.g. nitrogen loss in an Indian freshwater reservoir was enhanced by high methane concentrations[59].

We have compelling genomic evidence for denitrification potential within the Lake Zug MOB community, as denitrification genes (*narGHJI*, *napAB*, *nirKS*, and *norBC*) were found in the majority of *pmoA*-containing bins affiliated with the Methylococcales order (Fig. 5b). Of these, bin 4 (here classified as UBA4132 clade) closely affiliates with *Crenothrix* sp., which were shown to be capable of methane-dependent growth in anoxic, denitrifying conditions previously[42]. Notably, bins 5, 8, and 10 that belonged to the *Methylobacter* A clade also contained a combination of three different denitrifying enzymes. Members of the *Methylobacter* genus have been shown to encode genes for partial denitrification previously[31,57,60]. Interestingly, even though many gamma-MOB possessed and expressed genes related to denitrification (as well as fermentation-based methanotrophy), methane-dependent growth under apparent anoxia could only be observed for one discrete MOB morphotype (i.e. large rods). More cultivation-based as well as cultivation-independent studies are still needed to better understand the prevalence of anaerobic

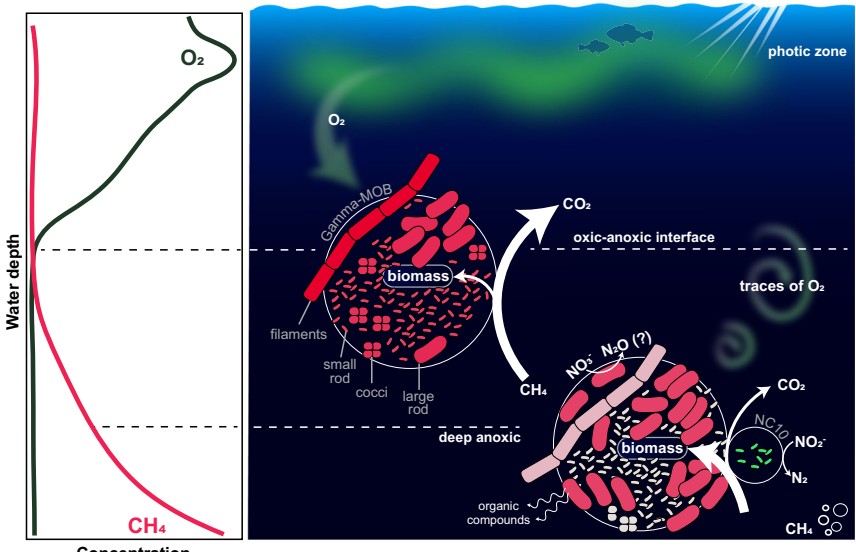

**Fig. 6 | Schematic overview of microbial methane oxidation in the anoxic hypolimnion of Lake Zug.** The left panel depicts representative water column profiles of oxygen and methane. The right panel illustrates the MOB groups, comprising gammaproteobacterial MOB (filaments, large rods, small rods, cocci) as well as anaerobic NC10 bacteria and their respective contribution to methane oxidation and assimilation at and below the oxic–anoxic interface. Note that the abundance, activity, and cell size of the different gamma-MOB groups are reflected by their respective numbers, color shading, and size in the figure. In the presence of oxygen, all four distinct Methylococcales-related gamma-MOB groups showed methane-dependent growth at comparable rates, whereas under apparent anoxia, only one group, the large rod-shaped gamma-MOB, was persistently active. Based on this, we conclude that under hypoxic conditions, all identified MOB groups contribute to methane oxidation, while large rod-shaped MOBs, as well as presumably NC10 bacteria, are responsible for the observed methane oxidation under anoxic conditions. Under anoxia, the assimilation of methane carbon into biomass exceeded the oxidation to carbon dioxide in this sampling campaign (see also Fig. S1). We envision that aerobic methane oxidation by gamma-MOB under apparent anoxia can be sustained by traces of oxygen that are periodically mixed into the anoxic waters (green arrows), in combination with anaerobic processes such as denitrification and/or fermentation that allow for energy conservation independently of oxygen. This image was created with Adobe Illustrator.

metabolisms of this environmentally relevant group of microorganisms.

The measured denitrification rates (to $^{30}N_2$) in our anoxic, nitrate-supplied incubations (Fig. 1d) were high enough to support methane oxidation at both anoxic depths, assuming a stoichiometric ratio of 5:8 for methane (oxidation to $CO_2$) and nitrate (reduction to $N_2$)[61]. However, no direct link between denitrification and methane oxidation rates could be shown experimentally, likely due to the large contribution of heterotrophic organisms to denitrification. It should be noted that to date, *Ca*. Methylomirabilis spp. are the only methanotrophs, which have been shown to denitrify all the way to $N_2$[62]. Methane-oxidizing gammaproteobacteria, on the other hand, notoriously lack the capacity to reduce $N_2O$ further to $N_2$, and the *nosZ* gene encoding for $N_2O$ reductase was consistently absent from our MOB bins. As such, the production of labeled $N_2O$ (both as $^{45}N_2O$ and $^{46}N_2O$) in our incubations (Fig. S4) might include a contribution from methane-oxidizing bacteria. The non-linear nature of the $N_2O$ accumulation throughout the experiment is indicative of rapid turnover of the produced $N_2O$ to $N_2$ by $N_2O$-reducing (non-methanotrophic) bacteria.

The sustained activity of *Methylobacter*-like MOB under anoxia indicates that these microorganisms efficiently remove methane not only at oxic-anoxic interfaces but also at anoxic water depths that contain nitrate and might periodically receive oxygen. Interestingly, the calculated per-cell methane oxidation rates of *Methylobacter*-like MOB under anoxic conditions (ca. 9.1 fmol $CH_4$ cell$^{-1}$ day$^{-1}$) (Supplementary Note 6) exceed the reported cellular rates of 'true' anaerobic archaeal and bacterial methanotrophs (ANME groups and NC10) making the 'aerobic' MOB very efficient 'anaerobic' methane oxidizers. This ecological success may be more pronounced in perturbation-prone anoxic water columns compared to e g. diffusion-driven anoxic and sulfidic sediments, as facultative aerobic bacterial methane oxidizers, may better cope with dynamic environments. Collectively, our findings highlight the need for a deeper understanding of the ecological role of gamma-MOB in anoxic environments.

Recently, the importance of aerobic methanotrophy for lacustrine carbon cycling was documented[63] and our results now accentuate the importance of this process also under anoxic conditions. With the increasing occurrence of seasonal or permanent anoxia in lakes, the importance of methanotrophy in methane removal within lacustrine systems can be expected to intensify, thus making a significant contribution to greenhouse gas mitigation and carbon storage in future scenarios.

## Methods

### Sampling and chemical profiling

Lake Zug is located in central Switzerland at an elevation of about 400 m above sea level. Water column profiling of Lake Zug was done in September 2017 at the deepest point of the lake (ca. 196 m water depth) in the northern basin (47°05'39.1"N 8°28'56.3"E). First, dissolved oxygen in the water column was recorded with normal and trace oxygen optodes (types PSt1 and TOS7, Presens) with detection limits of 125 and 20 nM, respectively[53]. Oxygen profiles were analyzed directly on board to select for discrete sampling depths. Water samples were collected from between 115 and 180 m water depth at 5–10 m intervals using a profiling in situ analyzer (PIA)[53], a syringe sampler with twelve 60-ml syringes, to collect samples for determination of methane, nitrate, and nitrite concentrations.

For determination of methane concentrations, the water was transferred into 120 ml serum bottles, and care was taken to avoid intrusions of gas bubbles. Copper chloride was added to the serum bottles to stop microbial activity and the bottles were closed

headspace-free with rubber stoppers and aluminum crimps and were kept at room temperature until analysis. For determination of nitrate and nitrite concentrations, water was sterile-filtered through 0.2-µm syringe filters and kept frozen until analysis on a commercial QuAAtro Segmented Flow Analyzer (SEAL Analytical Inc.).

Lake water for incubations was sampled from three water depths comprising the oxic–anoxic interface (123 m), 10 m below that (135 m), and from the anoxic, methane-rich hypolimnion (160 m). The water for incubations was collected with 5-l Niskin bottles and directly transferred into 2-l Schott bottles avoiding air intrusion. The bottles were allowed to overflow and then closed headspace-free with rubber stoppers. The incubation bottles were kept cold (4 °C) and in the dark until the start of the experiment (~1 week after sampling).

From the same depths as used for the incubations, water was collected for catalyzed reporter deposition fluorescence in situ hybridization (CARD FISH). For that, water samples were directly fixed with 2% paraformaldehyde and kept cold until further processing. Within 12 h after starting the fixation, the water from each depth was filtered onto polycarbonate filters (0.2-µm pore size; 25 mm diameter; Isopore™, Merck Millipore) at volumes of 10 and 20 ml, rinsed with sterile-filtered lake water and stored at −20 °C until analysis.

Samples for DNA and RNA extraction were collected with 5-l Niskin bottles from 123, 135, and 160 m. From each water depth, 1 l was filtered for DNA and RNA separately onto 0.22-µm Sterivex filter cartridges (Merck Millipore), directly onboard our boat using a peristaltic pump. Filters were stored at −20 °C for 2 months until DNA and RNA extraction.

In two consecutive years, sampling was conducted in the same manner, with absolute sampling depths varying due to the different depths of the oxic–anoxic interface. In October 2018, the oxic–anoxic interface was located at ca. 160 m, and in May 2019, the oxic-anoxic interface was located at ca. 175 m. In October 2018, we used a conductivity, temperature, and pressure profiler (CTD60, Sea&Sun Technology) equipped with a Clark-type oxygen sensor (accuracy ± 3%, resolution 0.1%) to record oxygen profiles. In May 2019, we used a CTD (CTD60, Sea&Sun Technology) equipped with an oxygen sensor (Sea-Bird) to record oxygen profiles. During both campaigns, discrete water column samples were collected with a 5-l Niskin bottle to determine concentrations of methane (not in May 2019), nitrate, and nitrite, as well as to collect water for incubation experiments.

## Methane concentration measurements

Methane concentrations were measured from discrete water samples (120 ml) by setting a 20-ml headspace with N$_2$ gas and injecting a headspace sample into a gas chromatograph (Agilent 6890N, Agilent Technologies), equipped with a Carboxen 1010 column and a flame ionization detector (accuracy of measurement ca. 6%).

## Calculation of methane fluxes

Methane fluxes were calculated for three zones in the water column that were defined based on changing slopes in the methane concentration profiles between 180 and 160 m water depth, between 160 and 135 m, and between 135 and 120 m. Methane fluxes were calculated according to Fick's first law following the equation $F = D \times \frac{dC}{dz}$, where $F$ is the vertical flux, $D$ is the turbulent diffusion coefficient, and $\frac{dC}{dz}$ is the concentration gradient of methane. While transport processes in the sediment are usually controlled by diffusion, the water column is influenced by internal currents and water movements. Therefore, we used a turbulent diffusion coefficient of 0.27 cm$^2$ s$^{-1}$ for the calculation[28].

## In situ CARD FISH-based cell counts

Gammaproteobacterial methane-oxidizing bacteria were targeted using probes Mγ705 (5′-CTG GTG TTC CTT CAG ATC-3′) and Mγ84 (5′-CCA CTC GTC AGC GCC CGA-3′)[64], following standard CARD FISH

procedures[65] with some probe-specific modifications. Briefly, cells were immobilized by embedding filter pieces in 0.1% low-gelling agarose. Endogenous peroxidases were inactivated with 0.01 M HCl for 10 min at room temperature, followed by permeabilization of cells with lysozyme (10 mg ml$^{-1}$ in 50 mM EDTA and 100 mM Tris−HCl for 30 min at 37 °C). Then, both probes (Mγ705 and Mγ84) were hybridized at 20% formamide concentration, followed by amplification with Alexa594-labeled tyramides at 46 °C for 25 min. Bacteria of the NC10 phylum related to *Candidatus* Methylomirabilis were visualized by CARD FISH using probe DBACT-1027[9]. Briefly, inactivation and permeabilization followed the same procedure as described above, followed by hybridization of HRP-labeled DBACT-1027 probe at 40% formamide and amplification with OregonGreen488-labeled tyramides at 46 °C for 25 min.

Filter sections were counterstained with 4′,6-diamidino−2-phenylindole (DAPI, 10 µg ml$^{-1}$ for 10 min at 4 °C) and viewed on an AxioImager Zeiss microscope. From these filter sections, we counted total cell numbers (based on DAPI signals), NC10 bacteria, as well as gamma-MOB numbers of morphologically distinct groups (appearing as small or large rods, cocci, and filaments), counting on average about 600, 200, and 250 cells per sample, respectively. Coccoid cell types encompassed free-living cells as well as cocci arranged in clusters. To determine the number of cocci in clusters, 30–70 randomly selected fields of view were screened, and the number of cells for each encountered cell cluster was determined.

## Stable isotope incubations

**Methane oxidation and denitrification rate measurements.** Methane oxidation activity was determined from lake water collected from 123, 135, and 160 m depths, amended with $^{13}$C-labeled methane. Incubation experiments were carried out under hypoxic conditions (with the addition of ca. 10 µM O$_2$) or anoxically (no O$_2$ added, amended with 20 µM $^{15}$N-labeled nitrate).

Lake water from the 2-l Schott bottles was placed into an anaerobic chamber (N$_2$/CO$_2$ atmosphere), and 220 ml were aliquoted into 250 ml serum bottles, leaving a circa 30 ml headspace. The bottles were then individually degassed with helium (Air Liquide Alphagaz 1) for about 15 min. Different incubations were set up in separate incubation bottles to investigate aerobic methane oxidation (referred to as hypoxic incubations) and anaerobic methane oxidation with nitrate as an electron acceptor (referred to as anoxic incubation). To each incubation bottle, 5 ml of pure $^{13}$C-labeled methane (99 atom% $^{13}$C, Sigma Aldrich, Cas. 6532-48-5) was added into the headspace and allowed to equilibrate with the water for about 1 h, resulting in a measured concentration of on average 50–90 µM $^{13}$C-methane dissolved in water (variability between treatments) and a $^{13}$C labeling percentage of >98%. To sample the gas from the cylinder, a gas mouse was installed in front of the cylinder along with a pressure regulator. The gas mouse was initially filled with water, which was then completely removed by opening the gas stream and allowing the gas flow to displace the water. After the gas mouse was completely filled with gas, a gas-tight glass syringe was used to sample the gas through a septum in the gas mouse and inject it into the incubation bottles through the rubber stopper. Gas loss during the transfer from the cylinder to the incubation bottle could have resulted in the observed concentration differences.

Hypoxic incubations were performed under low oxygen concentrations by adding 0.5 ml of pure oxygen into the headspace. This resulted in a final concentration of ca. 10 µM oxygen dissolved in water, but absolute concentrations varied between 3 and 50 µM between the different incubation setups and time points. Between the time points, we observed a reduction of oxygen concentration of several micromoles per liter, either due to microbial consumption or equilibration of oxygen between the water phase and the headspace. Therefore, hypoxic incubations were regularly replenished with oxygen by

injecting 0.5 ml of pure oxygen into the headspace. Oxygen concentrations in the anoxic incubations were below detection at all time points. The oxygen concentration was measured with oxygen sensor spots (Pyroscience) in the hypoxic (OXSP5 type, detection limit: 0.1% $O_2$ air saturation) and anoxic (TROXSP5 type, detection limit: 0.02% $O_2$ air saturation) incubations at all sampling time points. Sensor spots were glued to the inner side of the incubation bottles and read off with optical fibers from the outside.

Anoxic incubations were additionally supplied with $^{15}$N-labeled nitrate (Sigma Aldrich) to study denitrification. In order to reach a final concentration of 20 µM, ca. 50 µl of a degassed, 100-mM stock solution was added to the incubation, resulting in labeling percentage of about 50% (variable for the different incubation depths). Incubations were subsampled after 0, 1, 2, 5, 8 and 12 days; at each time point, subsamples of the incubation water and headspace were taken for methane oxidation and denitrification rate measurements, as well as for analysis of single-cell and bulk methane carbon assimilation. Subsamples from the incubation bottles were collected with a gas-tight glass syringe flushed with helium gas.

**Measurement and calculation of methane oxidation rates.** At each time point during the incubation, 6 ml of water was withdrawn from the incubation bottles and the volume was simultaneously replaced with helium. The sample was transferred into a 6 ml exetainer (Labco, UK), to which 50 µl saturated mercuric chloride solution was added to stop microbial activity. Methane oxidation rates were determined as the production of $^{13}CO_2$ from $^{13}$C-labeled methane. For the measurement, 3 ml of the sample was transferred from the original 6 ml exetainer into a 12 ml $N_2$-flushed exetainer and acidified with 100 µl of 20% phosphoric acid to release the dissolved $CO_2$ into the headspace.

Methane oxidation rates where determined by the increase in the ratio of $^{13}CO_2$/total $CO_2$ (in ppm) after the addition of $^{13}CH_4$ and quantified to mol l$^{-1}$ by DIC standards using a Picarro G2201-i cavity ring-down spectrometer with a Liaison A0301 interface, coupled to a AutoMate gas autosampler (Picarro Inc).

Rates were calculated from the slope of linear regression across the first five time points sampled during the incubations. An additional subsample was taken after 12 days. The measurements from this time point were, however, neglected for the calculations, as $^{13}CO_2$ production became exponential. The statistics on rate calculations are included in Supplementary Dataset 2.

**Measurement and calculation of denitrification rates.** Denitrification rates were determined as $^{30}N_2$ production from $^{15}$N-nitrate amended incubations by isotope ratio mass spectrometry (Isoprime Precision running ionOS v.4.04, coupled to an IsoFlow GHG sampling system, Elementar GmbH, Langenselbold, Germany). At each time point, a subsample of the headspace was taken by withdrawing 3 ml of the gaseous headspace and simultaneously replacing the removed volume with helium. The gas sample was transferred into a 12 ml exetainer (LabCo) that was pre-filled with helium-degassed water. Denitrification rates were calculated from the slope of the linear increase of $^{30}N_2$ in the headspace over the time course of the incubation. The rate of $^{30}N_2$ production was corrected for dilution of the headspace introduced by subsampling and by the measured labeling percentage of $^{15}$N-labeled nitrate. Labeling percentage of nitrate was determined using sulfamic acid (10 µl ml$^{-1}$ sample) and spongy cadmium (ca. 0.1 g ml$^{-1}$ sample) to convert nitrate to $N_2$. We also detected the production of $^{29}N_2$, but we did not include it in the calculation of denitrification rates due to the possibility of it being formed through N-transforming processes other than denitrification.

From the same water sample as used for methane oxidation measurements (see previous section), the production of $^{15}$N-labeled $N_2O$ from $^{15}$N-labeled nitrate was determined. The water sample was transferred from the original 6-ml exetainer into a 12-ml helium-degassed exetainer and spiked with 50 µl of $N_2O$ (3 µM final concentration). At all three incubation depths, we detected the production of labeled $N_2O$ (both as $^{45}N_2O$ and $^{46}N_2O$, Fig. S4). However, we observed a decrease in labeled $N_2O$ at later time points during the incubation. We attribute this decrease in $N_2O$ to microbial consumption because the incubation experiments were not performed with an unlabeled $N_2O$ pool to prevent consumption of the produced, labeled $N_2O$. Hence, we cannot derive volumetric $N_2O$ production rates from these experiments as the produced $N_2O$ might be readily reduced to $N_2$.

**Determination of bulk methane carbon assimilation.** At each time point of the incubation, 10 ml of the incubation water was sampled and filtered onto pre-combusted GF/F filters, air-dried, and frozen at −20 °C for storage. The samples were analyzed for C content and the respective isotopic composition by an elemental analyzer (Thermo Flash EA, 1112 Series) coupled to a continuous-flow isotope ratio mass spectrometer (Delta Plus XP IRMS; Thermo Finnigan, Dreieich, Germany). Bulk methane carbon assimilation rates were calculated from the relative $^{13}$C enrichment of the biomass (in $^{13}$C at.%) and quantified by the total C content per sample (in µg) based on caffeine standards. Bulk methane assimilation rates were calculated from the slope of linear regression calculated across five time points as described for the rate calculation of methane oxidation to carbon dioxide.

### Single-cell analysis
**CARD FISH assay.** From the incubations, 6 ml of water was fixed in 2% paraformaldehyde for 1 h at room temperature and subsequently filtered on gold-coated GTTP filters (0.2 µm pore size, 25 mm diameter, Millipore) for nanoSIMS analysis. Filters were air-dried at room temperature and kept at −20 °C until further processing. Catalyzed reporter deposition fluorescence in situ hybridization (CARD FISH) was performed to determine cell numbers at each time point of the incubation and to determine single-cell $^{13}$C-labeled methane-derived carbon assimilation by nanoSIMS analysis. For determination of cell numbers, filter pieces were embedded in 0.1% low-gelling agarose, for nanoSIMS analysis filter pieces were not embedded to avoid interference of the agarose treatment with the nanoSIMS analyses. Apart from the embedding, the filter pieces were treated identically for both types of analysis. CARD FISH was performed as described above.

**Nanoscale secondary ion mass spectrometry (NanoSIMS).** Single-cell methane carbon assimilation was determined from the incorporation of $^{13}$C into the biomass of individual cells via nanoSIMS analysis using a nanoSIMS 50 l (CAMECA). For nanoSIMS analysis, round filter pieces were punched out from the original filter to fit the nanoSIMS sample holder. Hybridized cells were marked with a laser microdissection microscope (LMD 6000B, Leica), and high-resolution images of the marked regions were taken with a Zeiss fluorescence microscope to help identify the cells during subsequent analysis (AxioImager M2 Zeiss microscope). The large gammaproteobacterial MOB (>1 µm) were pre-sputtered with a Cs$^+$ beam of 300 pA before measurement. A 1.5 pA Cs$^+$ primary ion beam was used for the measurements. Each measurement was recorded at a raster size of 25 × 25 or 35 × 35 µm (NC10 bacteria of 10 × 10 µm raster size) with a resolution of 256 × 256 pixels and a dwelling time of 1 ms per pixel over 40 planes.

**Processing of raw nanoSIMS data.** Raw nanoSIMS data was processed using the Look@NanoSIMS software package (version 2018-02–18)[66] in Matlab (version 2017b). Single cells were identified based on the corresponding CARD FISH images and the sulfur signal obtained from the measurement, used to define the cells as regions of interest (ROI). For each ROI, ratios of $^{13}C/(^{12}C + ^{13}C)$ were derived and used to calculate single-cell $^{13}$C enrichment values. Excess $^{13}$C atomic percent (at.%) values for each ROI were calculated by subtracting the

natural abundance of $^{13}C/(^{12}C + ^{13}C)$ (ca. 1.1%) determined from three background measurements per field of view. From the same ROIs, excess $^{15}N$ at.% values were derived from ratios of $^{12}C^{15}N/(^{12}C^{14}N + ^{12}C^{15}N)$ minus the natural abundance (ca. 0.36%). ROIs with a poison error larger than 5% (in $^{13}C/^{12}C$ and $^{12}C^{15}N/^{12}C^{14}N$ ratios) were excluded from subsequent analysis. Single-cell $^{13}C$ enrichments were visualized in Matlab using a modified boxplot function.

**Calculations.** For each MOB group (small or large rods, cocci, and filaments), we determined the cell counts, the biovolume and the $^{13}C$ assimilation at three time points during the incubations (1, 2, and 5 days). We used these data to calculate single-cell $^{13}C$ assimilation rates. Cell counts of the large rod-shaped MOB were additionally obtained at the start of the experiment (T0) and after 8 days of incubation.

**Determination of cellular volume, carbon content, and $^{13}C$ carbon assimilation.** The biovolume (in $\mu m^3$) of the different MOB groups was determined from fluorescence images. For coccoid gamma-MOB, we measured the cell diameter assuming a spherical cell shape. For small and large rod-shaped gamma-MOB, we assumed a cylindrical cell body with a half-sphere on both sides. For filamentous MOB, we measured the length of the whole filament, assuming a cylindrical cell shape. The carbon content per cell was determined with the formula $C_{(fg)} = 197 \times V^{0.46}$ (in fg C $\mu m^{-3}$), where $V$ is the biovolume of the cells[67]. We chose to use this formula to calculate the carbon content of a single MOB, even though cells of their size were not used to derive the original carbon-to-volume relationship. However, the only other formula is based on eukaryotic cells[68] and therefore seemed unsuitable to use. To calculate the $^{13}C$ assimilation per cell (in fmol $^{13}C$ cell$^{-1}$), the carbon content per cell was multiplied with the excess ratio of $^{13}C/(^{12}C + ^{13}C)$ derived from the nanoSIMS measurements.

Because CARD-FISH treatment may involve isotopic dilution of $^{13}C/(^{13}C + ^{12}C)$ and $^{15}N/(^{15}N + ^{14}N)$ ratios[69], comparisons of $^{13}C$ assimilation between the various gamma-MOB groups may be imprecise, due to the fact that *Crenothrix*-like filaments were not stained by CARD FISH, unlike the unicellular gamma-MOB.

### Sequencing analysis

**DNA and RNA extraction.** DNA and RNA were extracted from filtered water samples taken in September 2017 from the same water depths as used for the incubation experiments. DNA was extracted with the DNeasy PowerSoil Kit (Qiagen) according to the manufacturer's instructions. For RNA extraction, filters were briefly rinsed with nuclease-free water and RNA was extracted using the PowerWater RNA isolation kit (MoBio Laboratories), according to the manufacturer's instructions, including removal of genomic DNA by DNase I digestion. DNA and RNA yield was quantified using the Qubit dsDNA HS or RNA HS Assay kits and the Qubit 2.0 Fluorometer (Invitrogen).

**Metagenome sequencing, assembly and binning.** Library preparation and sequencing were performed at the Max Planck Genome Center Cologne, Germany (https://mpgc.mpipz.mpg.de/home/). Datasets were trimmed using trim galore v0.6.5 (https://www.bioinformatics.babraham.ac.uk/projects/trim_galore/), a wrapper improving cutadapt[70]. Trimmed reads were co-assembled using MEGAHIT v1.2.9[71] and contigs were manually grouped into coarse clusters of ~15,000 contigs using crossplots of contig coverage and GC content, guided by contigs containing the *pmoA* gene. The coarse clusters were subsequently manually binned using anvi'o v7.1[72], also guided by contigs containing the *pmoA* gene. The resulting 14 bins were annotated using anvi'o v7.1, with gene calling by prodigal v2.6.3[73], and functional annotation using COG[74], KEGG[75] and Pfam[76] databases. Bin completeness and contamination were estimated using checkM v1.2.2[77], bin taxonomy was determined using GTDB-tk v2.2.6[78], and

abundance was estimated using coverM v0.6.1 (https://github.com/wwood/CoverM) using minimap2 v2.24[79].

Concatenated marker gene phylogenies of the bins and reference sequences from the genome taxonomy database (GTDB v207)[80] and the "GEM-OTU" set of the genomic catalog of Earth's microbiomes (GEM)[81] were calculated using IQtree v2.1.2[82], with automatic model selection using ModelFinder[83] and 1000 ultrafast bootstrap replicates using UFBoot2[84]. The phylogenies are based on a concatenated alignment of 71 bacterial marker genes extracted from the genomes using hmmer v3.3.2[85] with HMMs included with anvi'o v7.1. The extracted genes were aligned using muscle v3.8.1551[86] as implemented in anvi'o v7.1, followed by concatenation of alignments using anvi'o v7.1. Phylogenies were visualized in iTOL[87]. Annotated bins were manually searched for fermentation genes by name, based on nomenclature in the COG database. Full list of genes that was searched for is included in Supplementary Dataset 5.

**Microbial community profiling and phylogenetic analysis of 16S rRNA gene sequences.** Microbial community composition based on 16S rRNA gene sequences in raw metagenomes was determined using phyloFlash (v3.3b2)[88] and the SILVA database (release 138)[89]. The phyloflash raw data is included in Supplementary Dataset 4. Relative abundances as presented in Fig. 5a were estimated based on SSU rRNA reads assigned to bacteria and archaea. Near full-length 16S rRNA gene sequences taxonomically assigned to Methylococcales were assembled from metagenomic 16S rRNA gene sequences using SPAdes assembler (version 3.11.1)[36] as implemented in phyloFlash (Supplementary Dataset 3). For the 16S rRNA gene phylogeny, 471 reference sequences were obtained from the Silva RefNR database (release r138.1)[89] with the following settings: taxonomy Methylococcales, sequence length >1399 nucleotides (nt), sequence quality >90, pintail quality >90 and supplemented with 16S rRNA gene sequences from the Methylococcales genomes used in the concatenated marker gene phylogeny. Erroneously binned 16S rRNA gene sequences were manually removed from the dataset. Sequences were aligned using muscle (version 3.8.1551). The phylogeny was calculated using IQtree (version 2.2.2.7)[82], with the best model identified by ModelFinder[83], and 1000 ultrafast bootstraps using UFBoot2[84]. The 16S rRNA gene phylogeny was rooted in the 16S rRNA genes corresponding to the Methylococcaceae family. The tree is included as a zoomable PDF file as Supplementary Dataset 1. Blue leaf labels correspond to sequences retrieved from publicly available genomes, red leaf labels correspond to metagenome-assembled sequences from this study.

**Metatranscriptomic sequencing and analysis.** Metatranscriptomic sequencing was performed using the Illumina HiSeq3000 platform (Illumina). Library preparation and sequencing were performed by the Max Planck-Genome-centre Cologne, Germany (http://mpgc.mpipz.mpg.de/home/). Metatranscriptome sequencing read datasets were trimmed using the cutadapt[70] wrapper trim-galore (https://github.com/FelixKrueger/TrimGalore) with default settings and automatic adapter detection. The trimmed reads were competitively mapped against all gene sequences in the MOB and NC10 genome bins retrieved from our data using coverM (https://github.com/wwood/CoverM), with minimum identity of 95% and minimum length fraction of 80%, exporting the read counts per gene sequence. A gene was considered transcribed when ten reads in the transcriptome data matched the gene. Read counts matching protein coding genes were used for the calculation of transcript per million (TPM) values[90] for each genome bin separately (Supplementary Dataset 5).

**Statistics and reproducibility.** In Figs. 2a, 3 and 4, the DAPI and FISH fluorescence images, using probes Mγ705 and Mγ84, are representative of 103 recorded images of samples obtained from hypoxic or anoxic incubations at five different time points. FISH fluorescence

images in Fig. 2a, using probe DBACT-1027, is representative of 10 recorded images from anoxic incubations at two time points.

In Fig. 3, the nanoscale secondary ion mass spectrometry (nano-SIMS) images of the large rod-shaped MOB are representative of 62 measured cells recorded at five time points during anoxic incubation.

In Fig.4, nanoSIMS images of large rod-shaped MOB are representative of 7 measured cells recorded at one time point during anoxic incubation.

In Fig. S7, FISH fluorescence images using probe DBACT-1027 is representative of 6 images obtained from three independent environmental samples.

In Figs. S8 and S9, nanoSIMS images are representative of 49 measured fields of view of multiple MOB cells in anoxic incubations at five different time points.

### Reporting summary

Further information on research design is available in the Nature Portfolio Reporting Summary linked to this article.

## Data availability

The metagenomics and metatranscriptomic datasets generated in this study have been deposited in the NCBI database under Bioproject number PRJNA977988. Publicly available sequences used for phylogenetic tree construction as presented in Supplementary Dataset 1 can be found under their respective accession numbers at NCBI. Source data are provided with this paper.

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

## Acknowledgements

We wish to thank the Swiss Federal Institute of Aquatic Science and Technology (Eawag) for the use of its research facilities as well as Alois Zwyssig, Mathias Zimmermann and Karin Beck for their help during sampling. We also thank Cas Cornet, Nadine Rujanski, Gabriele Klock-gether, Bram Vekeman and Daniela Tienken for technical assistance and help with sampling, sample preparation and analysis. This study was funded by the Max Planck Society and Eawag.

## Author contributions

S.S. and J.M. designed research and performed experiments. S.S., P.F.H., C.J.S., J.S.G., and J.M. organized and carried out sampling and characterized environmental samples. S.S. and S.L. performed nano-SIMS analysis. S.S., P.F.H., C.J.S., and G.L. performed sample measurements and analyses. J.S.G. extracted DNA and RNA and analyzed 16S rRNA data together with S.S. DRS analyzed metagenomic and meta-transcriptomic datasets. S.S., M.M.M.K., and J.M. interpreted data. S.S. and J.M. wrote the manuscript with contributions from all authors.

## Funding

## Competing interests

The authors declare no competing interests.
