## [Peer Review File · Nature Communications]

Persistent activity of aerobic methane-oxidizing bacteria in anoxic lake waters due to metabolic versatilityReviewer #1 (Remarks to the Author):

The present study focused on methane oxidation by aerobic Methylobacter-like bacteria in anoxic environments. Using a combination of stable isotope labelling, single cell imaging mass spectrometry and metagenomics, the authors investigated the activity and growth of aerobic gammaproteobacterial methanotrophs identified in the water column of a permanently stratified lake. Based on the obtained data the authors shows that these aerobic methane oxidizing bacteria (MOB) are present in the deeper anoxic layers of the lake and suggest that these are capable of oxidizing methane anaerobically, possibly using fermentation or denitrification under anoxic conditions.

The study is well written, excellently structured and designed, and shows a comprehensive experimental approach and effort. The choice of complementary methods from single cells assimilation rates to metagenomic information is an inclusive approach. The experimental design and the results are clearly and efficiently presented. The results support the discussion and interpretation of the data up to a certain point. The study is valuable and opens new research questions on the role of aerobic bacteria to methane oxidation in fresh water anoxic systems and C cycling. However, there are a series of concerns, detailed below, regarding data interpretation and main conclusions.

The main concern is that there is not enough experimental evidence to support "fermentation" as mechanisms employed by MOB in anoxic stratified layers, a term, which in my opinion is wrongly used here. One can envision denitrification and use of nitrate as electron acceptor, when oxygen is depleted, coupled to methane oxidation by MOB. It is not clear how these organisms will ferment methane, a strongly reduced compound. The proofs brought in are mostly circumstantial. Some overstatements are made, which need toning down, (below comments).

General comments:

Regarding data presented in Figure 1 and its interpretation there are couple of concerns:

If anoxic methane oxidation by large MOB is happening with nitrate as suggested by the authors why the MOB abundances decrease below 160m where there is a clear peak in nitrate and methane is not limited? How do the authors explain the similar nitrification rates (fig 1d, green bars) at 2 different nitrate concentrations?

Why was nitrification not measured/calculated below 160 m where there are comparable or higher nitrate concentrations to those measured at 135-160 m?

Specific comments:

Line 80: "Nitrate showed a steady decrease with depth.." What about the high peak around 170m depth?

Lines 102-105 and line 218: what about disruption of chemical gradients during sampling? If these cells function best at hypoxic conditions, sampling could have contributed to changes in oxygen, sulfide, methane, nitrate gradients etc. As the only possible explanation the authors go for "low in situ abundance".

Lines 203-208: do you have an explanation for lower ^{15}N enrichment vs ^{13}C enrichment in the active MOB, large rod-shaped cells? Is there an uncoupling of N and C assimilation or is there a limiting factor during incubations that can be brought into the discussion?

Line 255-257: if the cells are metabolically highly active, why not dividing and keeping low cell abundances? Authors suggest predation to keep the population reduced. Have the authors observed any type of ciliate during the microscopy investigations. Would the predation not affect all three groups of MOB detected in a similar manner?

Line 277-278: have such volatile fatty acids been measured?

Line 271-275: how a methane-based fermentation reaction should look like?

Lines 272-273: as the MOB contains monooxygenases, they need oxygen and therefore one cannot call it "fermentation", a process which involves reactions taking place anaerobiosis.

Line 279-282: unclear, rephrasing is necessary.

Lines 282-284: What would be the electron acceptor for the proposed fermentation process? How methane can be fermented? What are the specific/first indications the authors refer to here?

Lines 313-314: that is an overstatement considering that the authors cannot exclude traces of oxygen being present in their incubation experiments or during in situ sampling/measurements. Besides, methane monooxygenases are oxygen dependent, and these microorganisms have them. They simply do not possess the metabolic makeup to live, thrive and oxidize methane in strictly anaerobic conditions. If the authors can identify metabolites that support anaerobic oxidation by MOB that will be direct strong evidence supporting the current claims.

Reviewer #2 (Remarks to the Author):

This study is an important and interesting addition to the previous and ongoing studies on the role of aerobic gammaproteobacterial methanotrophs (Methylococcales) in consuming methane in hypoxic and even in anoxic conditions. The study is very well conducted with state-of-the-art methodology (following ¹³C-label from methane into CO₂ and bulk biomass and even into different types of single cells of Methylococcales; plus metagenomic data) and clearly reported and provides also the important experiment and dataset details as supplements. However, I have some concerns over whether the study is actually novel and substantial enough to be published in the journal. Furthermore, I have some additional minor comments. See my comments below

Major comments:

Major Comment 1:

The authors show with nanoSIMS+FISH that Methylococcales cells collected from hypoxic and anoxic water column of the study lake assimilate methane in hypoxic and anoxic conditions. This data is central to the study. I would like to know, what is here the true novelty compared to the previous excellent studies conducted by the research group, like Oswald et al. 2016:

<https://aslopubs.onlinelibrary.wiley.com/doi/full/10.1002/lno.10312>

where similarly the methane assimilation of single cells of Methylococcales of the study lake were shown in hypoxic and anoxic experimental conditions?

Is the novelty of the study in the suggestion of the proposed mechanisms of how the methane is consumed by the Methylococcales in hypoxic and anoxic conditions? Namely, in the study, authors also suggest that the methane oxidation by Methylococcales in hypoxic and anoxic conditions is coupled with fermentation and/or denitrification. But, authors do not show any direct proof that these processes actually are carried out by the Methylococcales in the study lake. They show that denitrification (to N₂ and N₂O) is happening in the lake but such kind of bulk process data cannot be linked specifically to Methylococcales. They also show via analyses of genomic bins that the Methylococcales in the study lake have genetic potential to denitrify, but that is still no proof that they would actually be active in denitrifying. Authors do very well cite papers where the fermentation and denitrification of Methylococcales is confirmed via process data and gene expression data, but in those papers the work has been done with non-lake Methylococcales isolates. Authors could find some use from the recent study by Khanongnuch et al. (2022), <https://www.nature.com/articles/s43705-022-00172-x#Sec17>, where the authors have actually isolated a representative of *Methylobacter* spp. of lake ecosystems and confirmed in laboratory conditions that it can drive a fermentation-type metabolism by converting methane into organic acids. But, even that paper (done in optimum laboratory conditions and not actually providing any gene expression data) cannot prove that Methylococcales in the study lake would actively ferment or denitrify.

To me it seems that actually to show that the study lake Methylococcales drive fermentation or denitrification, there is a need for metatranscriptomic or metaproteomic (gene expression) data,

from in situ samples and/or preferably from comparative experiments (oxic vs. hypoxic vs. anoxic), where the expression of fermentation and denitrification genes by Methylococcales of the study lakes is shown. Based on previous studies, metatranscriptomic study techniques are included in the research group's toolbox:

<https://ami-journals.onlinelibrary.wiley.com/doi/10.1111/1462-2920.14285>

If no new experiment can be done, can these older datasets be re-analysed in order to show the gene expression of Methylococcales of the study lake to fermentation and denitrification?

To show fermentation by Methylococcales, it would also be cool to show that the ^{13}C -label of methane goes into organic acids (via compound specific isotope analyses, LC-IRSM) but that might be challenging as organic acids are expected to be readily consumed in mixed microbial communities.

Major Comment 2:

Whether or not Methylococcales need any oxygen to activate methane and drive methane oxidation is an interesting question. The methane monoxygenase enzyme is suggested to need oxygen to function. In introduction and discussion, authors provide explanations on processes that could provide oxygen to methanotrophs living in anoxic water layers, like episodic oxygen intrusion from oxic layers (shallow and deep lakes) and photosynthesis (shallow lakes). The authors also acknowledge that they cannot completely rule out the possibility that some trace oxygen contamination could have affected their experiments.

I would like to know if authors would regard the recent findings on dark oxygen production relevant for their study:

Some methanotrophs (alphaproteobacterial at least) produce methanobactins which when reacting with metals (like Fe^{3+}) drive a water splitting reaction which produces free oxygen that can support methane oxidation: <https://journals.asm.org/doi/10.1128/aem.00286-21>

Could it sustain methane oxidation in the incubations of this study? By adding nitrate (which is readily reduced to nitrite in denitrifying conditions) one could actually enhance abiotic oxidation of Fe^{2+} to Fe^{3+} (coupled with nitrite reduction). The produced Fe^{3+} could then enhance the mentioned water splitting process. I actually wonder if the results by Oswald et al. 2016 on the addition of Fe oxides in enhancing methane oxidation in anoxic water samples of the study lake could be actually due to the water splitting process?

Furthermore, some ammonium oxidizing archaea can produce free oxygen from oxides of nitrogen in anoxic conditions:

https://www.science.org/doi/10.1126/science.abe6733?url_ver=Z39.88-2003&rfr_id=ori:rid:crossref.org&rfr_dat=cr_pub 0pubmed

Could it sustain methane oxidation in the incubations of this study? By adding nitrate, one could as well enhance this process.

Minor comments:

Lines 59-66. What about recent data indicating that methane oxidation by Methylococcales could be also coupled with reduction of iron and organic EAs. See e.g., <https://pubs.acs.org/doi/full/10.1021/acs.estlett.0c00436>

Lines 86-89. I doubt if you can determine the methane consumption depths only using the concentration data? Would $\delta^{13}\text{C}$ of methane be a helpful addition to this? Furthermore, what is the accuracy of methane conc measurements? Were the concentrations shown in figure based on replicate measurements?

Fig 1. Is the 1b the hypoxic treatment? It is described as "aerobic", which is of course correct but for clarity it would be better mention also the word "hypoxic" here.

Lines 102-103, 108-110 and other relevant parts. It would be good to actually see the

accumulation data of $^{13}\text{C}\text{O}_2$ to confirm the finding that $^{13}\text{C}\text{O}_2$ production started after lag-phase in some treatment and without lag phase in other treatment. Can it be shown in supplementary figure?

Line 235-239. Indeed, strain *Methylobacter tundripaludum* SV96 is rod. But, the recent lake isolate by Khanongnuch et al. (2022), i.e., *Methylobacter* sp. S3L5C is actually cocci. And so is *Methylobacter psychrophilus* Z-0021, which is close relative to S3L5C. According to Khanongnuch et al. analyses, the strain S3L5C represents a large cluster of *Methylobacter* spp. present in lake water columns. Hence, *Methylobacter* in lake water columns can be also cocci. Can you be sure that the rods were all *Methylobacter*? Could there be additional analyses, like qPCR, to show that the cell number increase of rods is correlated with increase in *Methylobacter*?

Line 268-271. Actually, I think that Kalyuzhnaya et al. 2013 suggest that the fermentation mode leads to decreased cellular assimilation and CO_2 production and increased excretion of fermentation products, like VFAs.

line 407. Pure $^{13}\text{C}\text{-CH}_4$? Can you specify the ^{13}C -label percentage? Not completely 100%?

line 409. Did you measure the dissolved CH_4 conc in incubation bottles?

line 422. Did you include non- ^{13}C – labeled controls to assess the background change in $^{13}\text{C}\text{-CO}_2$? I assume that adding nitrate will anyway enhance production of CO_2 from organic matter oxidation. In that process, also natural $^{13}\text{C}\text{-CO}_2$ is produced. How did you determine which part of the increase in $^{13}\text{C}\text{-CO}_2$ is from your added ^{13}C - label and which part is increase in natural $^{13}\text{C}\text{-CO}_2$?

Line 423-425. How did you make sure that no unwanted O_2 contaminated the incubations during these samplings?

Line 436. Can you explain what is the notation " $^{13}\text{C}\text{O}_2/^{12}\text{C}\text{O}_2 + ^{13}\text{C}\text{O}_2$ (in ppm)"? Ratio of $^{13}\text{C}\text{O}_2$ to $^{12}\text{C}\text{O}_2$ plus concentration of $^{13}\text{C}\text{O}_2$?

Figure 4. Figure caption: "for 24 hours". Unclear, since the data in figure goes until 5 days.

Reviewer #3 (Remarks to the Author):

Summary

The authors investigated the assimilation and oxidation of ^{13}C -methane of aerobic methanotrophs from Lake Zug under suboxic and anoxic+nitrate in short-term incubations over 5 days. They measured an oxygen profile depth and sample between 115 m and 180 m at 5-10 m intervals for chemical parameters methane, nitrate and nitrite. Incubation and CARD-FISH samples were collected from three depths, 123 m, 135 m and 160 m, DNA was collected from 125m, 135m and 160m respectively. In 2018 and 2019 incubations were done from deeper samples ($\geq 160\text{m}$ -180m, 175m-190m respectively). In 2018 and 2019 chemical profiles, bulk methane oxidation and bulk assimilation under oxic and anoxic+nitrate conditions were done, but not DNA, denitrification, CARD-FISH or nanoSIMS analysis. The authors targeted aerobic methane-oxidizing Gammaproteobacteria with CARD-FISH. The targeted methanotrophs were categorized into small or large rods, cocci and filaments, and cluster-forming coccoid cells. The incubations were done with ^{13}C -methane under suboxic ($10\ \mu\text{M}\ \text{O}_2$) or anoxic (no O_2 added, but 15N -nitrate added) conditions for a total of 6 incubations per year. The authors monitored the O_2 concentration in the incubations (data not shown). Five timepoints over 8 days were taken for bulk $^{13}\text{C}\text{-CO}_2$ production and denitrification rates, as well as single cell analysis with nanoSIMS and bulk methane-C assimilation of filtered cells.

The authors found that large rods identified as Methylococcales by CARD-FISH continued to assimilate methane derived carbon under anoxic nitrate amended conditions at similar rates than under oxic conditions, which was also the case for some Crenothrix filaments (identified by morphology), but was not the case for cocci and small rods. The authors suggest that fermentation and denitrification may be used by these MOB, which are mechanisms that have been shown in previous studies to be oxygen saving mechanisms in aerobic methanotrophs, although for the initial methane oxidation oxygen was still needed in those experiments which also the authors acknowledge in line 65.

The authors reconstructed bins of Methylobacter, KS41, Methylovulum, UBA4132, UBA10906, SXIZ01 and Methyloirabilis an anaerobic methanotroph producing O₂ from nitrite which is then used to oxidize methane. All but KS41 seem to have nitrate reduction capability. Fermentation genes were not analyzed.

The authors convincingly show that large rods within Methylococcales continue to assimilate methane derived carbon under anoxic+nitrate conditions with very low oxygen concentrations. Therefore, either the large rods have substantial ability of saving oxygen by known (nitrate reduction, fermentation) or unknown mechanisms, produce their own oxygen (which is not known for Methylococcales), or use fully or partially other electron acceptors (nitrate reduction or else), or other unknown mechanisms – what the mechanisms are remains a question to be answered.

Major comments

1) Methyloirabilis

In line 160 they mention that Methyloirabilis was found in the DNA data (a bin was recovered), which is not surprising given that a subset of the authors has found abundant Methyloirabilis limnetica (up to 27%) in an earlier study of the same lake. This does complicate the interpretation of the following parameters: a) bulk ¹³C-CO₂ production which could come fully or partially from Methyloirabilis in the sample. b) Similarly, bulk ¹³C-methane assimilation might contain the anaerobic methane oxidizer Methyloirabilis. This seems likely as the anoxic incubations were amended with nitrate. c) ¹⁵N₂ production which is also possible to be formed by Methyloirabilis. It is somewhat surprising that the presence of Methyloirabilis is not discussed and shown throughout the manuscript.

I strongly suggest to include the potential role of Methyloirabilis throughout the manuscript which might change several conclusions for bulk methane oxidation rates and assimilation rates. Further Methyloirabilis should be included in the figures of the DNA results (e.g. Fig. 2a,bc and Fig. 5) and an abundance depth profile should be included analogous to aerobic MOB in Fig. 2c to be able to assess the abundance of this anaerobic methanotroph, which is currently not possible. In case Methyloirabilis has much lower abundance than Methylococcales still a potential contribution should be discussed.

Growth rates based on morphology and CARD-FISH

According to S3 the large rods did not increase their cell numbers over time, but rather stayed the same from day1 to day2 and then declined slightly to day5. See methods to clarify how the growth rates were calculated and potential issues.

2) Oxygen contamination

The authors monitored the oxygen in the incubations over time, which is not usually done but is very interesting important information, but currently the data is not shown. I suggest to show the data in the manuscript because it can rule out presence of substantial oxygen concentration, which often is difficult to avoid and e.g. taking timepoints, or oxygen contaminated gases (¹³C-CH₄, He) could introduce oxygen. The purity of the used gases, manufacturer and LOT nr (He, ¹³C-CH₄) should be stated including the contamination levels of oxygen if available, and how ¹³C-CH₄ was taken from the bottle without introducing oxygen. This information will be of interest for other researcher who would also like to study aerobic methanotrophs under oxygen starvation.

3) Morphological characterization and nanoSIMS result

The authors show one example of each category in Fig. 3a, b. The main shape "large rod" is only shown once and in my opinion, it would be important to show more than one image of such a cell. It is crucial to also show the CARD-FISH + nanoSIMS image of the anoxic+nitrate active cell and not just the oxygen+methane incubation images.

4) DNA results/Phylogenetic analysis

The title of the manuscript states "Methylobacter-like" aerobic methanotrophic bacteria for the "large rod shape" Methylococcales. I think the taxonomic identification "Methylobacter-like" is not

justified based on the data presented. a) multiple taxonomic groups within Methylococcales are present including several uncultivated genera (6 genera Methylobacter_A (3bins) SXIZ01(1bin) KS41 (3bins), Methylovulum (1bin), UBA4132 (2bins) and UBA10906 (3 bins)) making it impossible to identify which one is the "large rod". In my opinion shape is not a good indicator in here especially since the shape for several uncultivated groups is unknown b) for identifying the organism further analysis e.g. a targeted FISH probe would be needed. Therefore, I suggest removing the conclusion Methylobacter throughout the manuscript.

Figure 2: The phylogenetic analysis of the aerobic methanotrophs present is very important, since the claim is that one "group" but not all aerobic methanotrophs are capable of some continued anaerobic activity. The 16S rRNA tree in Fig. S2 does not suggest that e.g. GammaMOB1 160m or 123m is Methylobacter since the clades are not significantly. The lacustrine Crenothrix is supported and the CAB2E06 is supported too. There are sequences in 3 additional clades without a taxonomic identification.

In Figure 2b 6 methanotrophic groups are shown. Methylobacter is not a valid aggregation based on tree S2. Methylovulum in that figure does not occur in the tree S2, neither does Methyloglobulus which is surprising. Further, it is unclear which sequences are included in "uncultured" vs "other Methylococcales". I suggest using the classification of the tree instead, supported are "Crenothrix" and "CAB2E06". E.g. by looking at significance lower than 90% like 80 or 70% in the tree together with % identity between sequences a better classification can be found (general classifiers sometimes perform poorly on genus level and a tree should be made). Since it is unclear if the "anoxic" trait is at strain/species/genus level, and there are only about 10 16S rRNA sequences showing the depth distribution of all 10 sequence variants instead of the grouping in Fig. 2b) would be very useful. In case the bins have 16S sequences it could be tried to connect the classifications.

Minor comments

Titel and Abstract, please see major points.

Introduction

Line 30-32 substantiate with citations.

39 consider mentioning the proposed mechanism of Methylovulum

69 e.g. Methylovulum is an anaerobic bacterial methanotroph, consider formulating more specific

Results

Figure 1 b, c, please consider using same scale for aerobic and anaerobic MO for a better comparison.

Figure 1 e, showing small rods as x10 is unusual and makes it difficult to read. Consider showing all with same scaling or show the biovolume data, or two figures as a suggestion.

Figure 2 needs major revision see major points

Figure 3 please show all data points, the asterisk is not acceptable. I am surprised the authors consider Crenothrix under anoxic+nitrate condition an outlier since Oswald et al 2017 has shown a similar result for Crenothrix from Lake Zug.

114 discuss further in discussion section, please clarify why a correlation was expected? It seems aerobic methanotrophs do not possess nosZ usually. Were the authors trying to measure Methylovulum activity 30N2? Please consider discussing also non methanotrophs as N2 producers.

115 The N2O production is interesting please consider showing the data.

122 Please compare to literature values. Please consider that production of dissolved organic carbon is not included here (e.g. formate, acetate, methanol, formaldehyde, lactate..) which might play a role.

137 it seems the increase in abundance with depth here is not as pronounced as in the DNA data. Consider to discuss.

142 Because Crenothrix was not identified with a probe either separately or in the mix, it cannot be entirely excluded that the low activity cells under anoxic conditions may not be Crenothrix. Please add to discussion.

156 see major comments on Methylobacter

167 since fermentation is a big discussion point, consider analyzing potential pathways

184 did you consider sequencing the bottle, it might give an indication which methanotroph was

growing.

186 according to Table S3 the cell counts of large rods did not increase, since no standard deviations are given I would expect to see an increasing trend from day1 to day5 which seems not to be the case. Especially because this is one of the main results. Please include day 0 in Tables S3 so a comparison can be made.

190 did you see the large rods after 8 days as well?

200 consider moving this to discussion, since no oxygen reductant was added residual oxygen is an explanation too

Discussion

Please consider incorporating more examples of aerobic methanotrophs in anoxic habitats. This is observed quite often and has been studied in some other lakes and other environments e.g. marine too. E.g. Su, G., Lehmann, M.F., Tischer, J. et al. Water column dynamics control nitrite-dependent anaerobic methane oxidation by *Candidatus "Methylomirabilis"* in stratified lake basins. *ISME J* 17, 693–702 (2023). <https://doi.org/10.1038/s41396-023-01382-4>

220 the bathymetry of a lake can also influence the methane profile.

228 Figure 1e only shows aerobic methanotrophs, *Methylomirabilis* or *Methanoperedens* are not shown in the Figure. Please add *Methylomirabilis* and check if other archaeal methanotrophs are present.

245 please clarify somewhere why nitrate was added to anoxic incubations

236 see major point "*Methylobacter*"

252 please cite and compare to finding of Oswald, K., Graf, J., Littmann, S. et al. *Crenothrix* are major methane consumers in stratified lakes. *ISME J* 11, 2124–2140 (2017).

<https://doi.org/10.1038/ismej.2017.77>

who also showed activity of *Crenothrix* under anoxic+nitrate conditions in lake Zug.

Note 3: 13 bins are shown in Fig.2, in Note 3 14 are mentioned, please clarify

255 the authors do not quantify *Methylomirabilis* here

276 fermentation by methanotrophs may occur, please phrase more carefully. Fermentation by methanotrophs in Lake Zug was not shown. The mechanism in my opinion is unknown, it could be trace amount of oxygen, nitrate reduction or fermentation or an as yet unknown mechanism.

278 volatile fatty acids have not been mentioned before in the manuscript, please include in introduction and explain in detail including references or consider leaving it out.

283 Consider revising the conclusion that fermentation has great relevance. The experiments done in this study do not allow such a conclusion, but the authors incubated with additional nitrate and showed N₂ and N₂O production which are interesting results and N₂O production could be discussed instead.

286ff Please mention that this is also the case under oxic conditions.

297 please consider also discussing N₂O production, since N₂ production is not expected from *Methylococcales*

Methods

364 consider showing a conductivity profile since it is interesting to see the stratification.

393 are cluster forming coccoid cells and cocci the same? I only see cocci in all figures, consider adding the dimensions of the cells also here.

401 please add how much volume was added to reach 20 uM nitrate and please clarify why nitrate was added in the text.

402 please add manufacturer and purity for all gases including the N₂/CO₂ mix, He and 13C-methane. What kind of O₂ scrubber was used? Please clarify in text.

409 the concentration in water seems too low considering 5ml CH₄ in a 30ml headspace. Please clarify. Where does the variability originate from? Please clarify in text.

411 in the supplements 20 uM are stated. Where does the variability come from? Please clarify in text.

416 oxygen was monitored see major point consider including the data

472 please clarify how many cells were counted to determine the numbers in table S3.

514

In my opinion using the start and end point here is only valid if there was exponential growth observed and expected, I do not think this kind of calculation is valid for the anoxic incubations, since Table S3 shows that there was no exponential growth but rather a stable to decline of cell numbers for all categories including the large rods. To me is unclear how a positive growth rate was calculated? Please provide the cell numbers from the beginning of the experiment in table S3.

Even if t_0 was lower, it looks like the large rods grew less and declined a bit towards day 5 and the exponential equation is probably not a good fit.

As far as I understand the incubation water was also stored for a week until incubation, and the I assume the Card-FISH cells were fixed and filtered immediately after sampling? Please clarify in the methods which cell number was used for t_0 in the calculation. If the value from immediately fixed cells was used, there is a chance the large rods grew in that period. So I think using day 1 and day 5 or 8 if available would be more suitable to evaluate what was growing under the respective conditions.

If the large rods did not increase their cell number, there is the question why the assimilation rates were so high nevertheless. I think that is something that should be discussed.

576 consider attaching the phyloflash result so an overview of the microbial community can be gained.

Response to the reviewers' comments

Reviewer #1 (Remarks to the Author):

The present study focused on methane oxidation by aerobic Methylobacter-like bacteria in anoxic environments. Using a combination of stable isotope labeling, single cell imaging mass spectrometry and metagenomics, the authors investigated the activity and growth of aerobic gammaproteobacterial methanotrophs identified in the water column of a permanently stratified lake. Based on the obtained data the authors shows that these aerobic methane oxidizing bacteria (MOB) are present in the deeper anoxic layers of the lake and suggest that these are capable of oxidizing methane anaerobically, possibly using fermentation or denitrification under anoxic conditions.

The study is well written, excellently structured and designed, and shows a comprehensive experimental approach and effort. The choice of complementary methods from single cells assimilation rates to metagenomic information is an inclusive approach. The experimental design and the results are clearly and efficiently presented. The results support the discussion and interpretation of the data up to a certain point. The study is valuable and opens new research questions on the role of aerobic bacteria to methane oxidation in fresh water anoxic systems and C cycling. However, there are a series of concerns, detailed below, regarding data interpretation and main conclusions.

The main concern is that there is not enough experimental evidence to support “fermentation” as mechanisms employed by MOB in anoxic stratified layers, a term, which in my opinion is wrongly used here. One can envision denitrification and use of nitrate as electron acceptor, when oxygen is depleted, coupled to methane oxidation by MOB. It is not clear how these organisms will ferment methane, a strongly reduced compound. The proofs brought in are mostly circumstantial. Some overstatements are made, which need toning down, (below comments).

Indeed, we agree with the reviewer that methane cannot be fermented and we now refer in our manuscript to ‘fermentation-based methanotrophy’ instead. This term was originally used by Kalyuzhnaya et al. (<https://doi.org/10.1038/ncomms3785>) who showed that a pure culture of a *Methylomicrobium* strain converted, under oxygen-limiting conditions, methane to fatty acids and hydrogen through a combination of methane oxidation via formaldehyde to pyruvate and subsequently fermentation. The study also identified fermentation genes that were upregulated during this process.

We have now looked for the presence and expression of these genes *in situ* in Lake Zug. These new results show presence and expression of fermentation genes by MOB in the anoxic hypolimnion. Together with the high expression of methane monooxygenase these results now directly link fermentation-based methanotrophy to MOB *in situ*. To be clear, we do assume that traces of oxygen

needed for the first step of methane oxidation are present even in these apparently anoxic waters. However, by employing a fermentation pathway, oxygen would not be required for respiration. This is analogous to the proposed mechanism behind denitrification by MOB where nitrate is thought to replace oxygen as an electron acceptor but oxygen is still required for the initial step of methane oxidation. Both mechanisms are thus in principle similar in that they explain how these aerobic MOB can 'save' oxygen and thrive in the presence of nearly non-detectable amounts of oxygen.

We have now added new metagenomics and metatranscriptomic data in support of fermentation-based methanotrophy in Lake Zug. Additionally, we have revised the manuscript to clarify the proposed mechanism behind this process based on previous literature (see lines 73-79).

General comments:

Regarding data presented in Figure 1 and its interpretation there are couple of concerns:

If anoxic methane oxidation by large MOB is happening with nitrate as suggested by the authors why the MOB abundances decrease below 160m where there is a clear peak in nitrate and methane is not limited? How do the authors explain the similar nitrification rates (fig 1d, green bars) at 2 different nitrate concentrations?

We can only speculate about the observed decrease in MOB abundance. We think that this could be due to reduced oxygen transport to these depths as downward oxygen transport by eddies decreases with increasing distance to the oxycline. As mentioned above, both pathways (denitrification and fermentation-based methanotrophy) still require oxygen for the first step of methane oxidation. We now add this consideration to Supplementary Note 5 (line 111).

Regarding denitrification rates, these often do not correlate with *in situ* nitrate concentrations. Denitrification consumes nitrate and therefore highest rates are often observed where nitrate concentrations are lowest. This is in agreement with the lowest rate measured at the highest nitrate concentration at 123 m.

Why was nitrification not measured/calculated below 160 m where there are comparable or higher nitrate concentrations to those measured at 135-160 m?

During sampling, only oxygen profiles are recorded in real time, on the basis of which the sampling depths for the incubations are selected. Our aim was to sample from the oxic-anoxic interface, the chemocline, and from an anoxic, methane-rich depth. Nitrate concentrations were measured only after returning to our home laboratory. Thus, the sampling depths were selected without knowing the exact nitrate concentrations.

Specific comments:

Line 80: “Nitrate showed a steady decrease with depth..” What about the high peak around 170m depth?

With the exception of this one sample, all other sampled depths show a continuous nitrate concentration decrease with depth, which is representative of how nitrate profiles typically look like in this lake. We have now changed the text to ‘nitrate concentrations generally decreased with depth’ (line 95).

Lines 102-105 and line 218: what about disruption of chemical gradients during sampling? If these cells function best at hypoxic conditions, sampling could have contributed to changes in oxygen, sulfide, methane, nitrate gradients etc. As the only possible explanation the authors go for “low in situ abundance”.

The samples were carefully retrieved with Niskin bottles, which is a well-established sampling method in aquatic sciences. It is commonly used to assess the spatial distribution of dissolved gasses, nutrients, and other parameters, even at higher spatial resolutions.

We consider it very likely that the initial lag phase observed in incubations from the oxic-anoxic interface may be attributed to low activity of the methane-oxidizing bacteria (MOB) *in situ*, as methane concentrations were non-detectable at the oxycline depth at the time of sampling. In contrast, in our incubations, methane was supplied at a concentration of 50 to 90 μM . Thus, it is very likely that the MOB simply needed time to start expressing all the enzymes needed to grow and oxidize methane under the incubation conditions.

Lines 203-208: do you have an explanation for lower ^{15}N enrichment vs ^{13}C enrichment in the active MOB, large rod-shaped cells? Is there an uncoupling of N and C assimilation or is there a limiting factor during incubations that can be brought into the discussion?

The low uptake of ^{15}N from added ^{15}N -nitrate is probably due to nitrate not being the only source of nitrogen for these cells, which also (and likely preferentially) assimilate ammonium (<https://doi.org/10.1038/s43705-022-00172-x>). Non-labeled ammonium is present at high concentrations (up to 20 μM) in the *in situ* water and hence also in our incubations. We now point this out in the revised manuscript (lines 205).

Line 255-257: if the cells are metabolically highly active, why not dividing and keeping low cell abundances? Authors suggest predation to keep the population reduced. Have the authors observed any type of ciliate during the microscopy investigations. Would the predation not affect all three groups of MOB detected in a similar manner?

Highly active cells that assimilate carbon but don't divide would have to substantially increase their biovolume, which was not apparent in our samples. However, we have now added cell counts after 8 days of incubation which show that the cell numbers did increase eventually (see lines 169).

Regarding the effect of grazers on the different MOB morphotypes - grazers are known to have size preferences for their prey and we know that bacterivorous anaerobic protists are present at these depths in the lake, and that they even prey on *Methylobacter*-like MOB (<https://doi.org/10.1038/s41586-021-03297-6>). Additionally, viral lysis can contribute to loss of MOB cells in the incubations.

Line 277-278: have such volatile fatty acids been measured?

We performed analyses on our samples to identify methane-derived organic compounds that were formed during our incubations, using LC-MS. However, we could not detect them, possibly due to the rapid consumption of such organic acids by the microbial community. Indeed, our nanoSIMS data showed uptake of methane-derived organic carbon by other, presumably non-methanotrophic bacteria (see for example new figures S8 and S9).

Line 271-275: how a methane-based fermentation reaction should look like?

Please see the response to the first comment.

Lines 272-273: as the MOB contains monooxygenases, they need oxygen and therefore one cannot call it "fermentation", a process which involves reactions taking place anaerobiosis.

We agree, please see the explanation above.

Line 279-282: unclear, rephrasing is necessary.

We revised the sentence (line 355).

Lines 282-284: What would be the electron acceptor for the proposed fermentation process? How methane can be fermented? What are the specific/first indications the authors refer to here?

Please see explanation above. We have now included a better overview of prior literature regarding fermentation-based methanotrophy in the revised manuscript (line 73-79).

Lines 313-314: that is an overstatement considering that the authors cannot exclude traces of oxygen being present in their incubation experiments or during in situ sampling/measurements. Besides, methane monooxygenases are oxygen dependent, and these microorganisms have them.

They simply do not possess the metabolic makeup to live, thrive and oxidize methane in strictly anaerobic conditions. If the authors can identify metabolites that support anaerobic oxidation by MOB that will be direct strong evidence supporting the current claims.

We acknowledge that MOB presumably need oxygen for the first step of methane oxidation and we can imagine that oxygen can in fact be available in trace amounts *in situ* as well as in our incubations. We now clarify this in the revised manuscript. Nonetheless, we cannot detect this oxygen using state-of-art oxygen sensors, making the observed methane oxidation by MOB effectively anaerobic. Based on the lack of detectable oxygen (as well as the high measured denitrification rates) this system as well as our incubations are definitely ‘anoxic’. However, this is not to be compared to e.g. anoxic marine sediments that are typically also sulfidic and thus have low redox potentials. We do not attempt to discuss activity of bacterial MOB under such ‘strictly’ anaerobic/low redox conditions as such conditions are not relevant for our Lake Zug water column.

Nonetheless, we have now down-toned the respective statement from ‘the most efficient’ to ‘very efficient’ (line 399) and we have added an additional discussion in the manuscript about the role of oxygen *in situ* as well as in the incubations.

Reviewer #2 (Remarks to the Author):

This study is an important and interesting addition to the previous and ongoing studies on the role of aerobic gammaproteobacterial methanotrophs (Methylococcales) in consuming methane in hypoxic and even in anoxic conditions. The study is very well conducted with state-of-the-art methodology (following ¹³C-label from methane into CO₂ and bulk biomass and even into different types of single cells of Methylococcales; plus metagenomic data) and clearly reported and provides also the important experiment and dataset details as supplements. However, I have some concerns over whether the study is actually novel and substantial enough to be published in the journal. Furthermore, I have some additional minor comments. See my comments below.

Major comments:

Major Comment 1:

The authors show with nanoSIMS+FISH that Methylococcales cells collected from hypoxic and anoxic water column of the study lake assimilate methane in hypoxic and anoxic conditions. This data is central to the study. I would like to know, what is here the true novelty compared to the previous excellent studies conducted by the research group, like Oswald et al. 2016: <https://aslopubs.onlinelibrary.wiley.com/doi/full/10.1002/lno.10312>

where similarly the methane assimilation of single cells of Methylococcales of the study lake were shown in hypoxic and anoxic experimental conditions?

Is the novelty of the study in the suggestion of the proposed mechanisms of how the methane is consumed by the Methylococcales in hypoxic and anoxic conditions?

We recognize that we did not do a good job highlighting what distinguishes this manuscript from our previous work. In our previous work we looked at the NC10 bacteria (Graf et al., EMI, 2018), *Crenothrix* (Oswald et al., 2017) or gammaproteobacterial MOB (Oswald et al., 2016, L&O). In this manuscript we look at yet another MOB, namely the Methylobacter-like ‘large rods’. We kept seeing high abundances of these rods in anoxic waters, suggesting that they were active; however, we nor others have investigated their activity previously. Therefore, we designed a study to investigate their role in lacustrine methane oxidation. We compare their activity to the organisms we previously studied (such as *Crenothrix*) to get a perspective on their importance in ‘anaerobic’ methane oxidation. Our main new finding is that the Methylobacter-like large rods are in fact by far the most important methane oxidizers in the anoxic lake waters and that their activity is hardly diminished under anoxia. Furthermore, we show that under anoxic conditions, disproportionate amounts of methane carbon were assimilated into microbial biomass, presumably through a little-investigated process of fermentation-based methanotrophy, implying that a large fraction of methane carbon may be retained in anoxic basins due to MOB activity. Thus we show how fermentation-based methanotrophy, a mechanism described for pure MOB cultures, might contribute to their ecological success. We have now better highlighted the comparisons to our earlier studies in the revised manuscript.

Namely, in the study, authors also suggest that the methane oxidation by Methylococcales in hypoxic and anoxic conditions is coupled with fermentation and/or denitrification. But, authors do not show any direct proof that these processes actually are carried out by the Methylococcales in the study lake. They show that denitrification (to N₂ and N₂O) is happening in the lake but such kind of bulk process data cannot be linked specifically to Methylococcales. They also show via analyses of genomic bins that the Methylococcales in the study lake have genetic potential to denitrify, but that is still no proof that they would actually be active in denitrifying. Authors do very well cite papers where the fermentation and denitrification of Methylococcales is confirmed via process data and gene expression data, but in those papers the work has been done with non-lake Methylococcales isolates. Authors could find some use from the recent study by Khanongnuch et al. (2022), <https://www.nature.com/articles/s43705-022-00172-x#Sec17>, where the authors have actually isolated a representative of *Methylobacter* spp. of lake ecosystems and confirmed in laboratory conditions that it can drive a fermentation-type metabolism by converting methane into organic acids. But, even that paper (done in optimum laboratory conditions

and not actually providing any gene expression data) cannot prove that Methylococcales in the study lake would actively ferment or denitrify.

To me it seems that actually to show that the study lake Methylococcales drive fermentation or denitrification, there is a need for metatranscriptomic or metaproteomic (gene expression) data, from *in situ* samples and/or preferably from comparative experiments (oxic vs. hypoxic vs. anoxic), where the expression of fermentation and denitrification genes by Methylococcales of the study lakes is shown. Based on previous studies, metatranscriptomic study techniques are included in the research group's toolbox: <https://ami-journals.onlinelibrary.wiley.com/doi/10.1111/1462-2920.14285>

If no new experiment can be done, can these older datasets be re-analysed in order to show the gene expression of Methylococcales of the study lake to fermentation and denitrification?

We agree with the reviewer's suggestion. We have now looked for expression of fermentation as well as denitrification genes in our *in situ* metatranscriptomes that were obtained from all three investigated depths. These data are now included in the revised Figure 5b.

We could successfully detect expression of a particulate methane monooxygenase, as well as numerous denitrification and fermentation genes by the MOB, including the 3 bins belonging to the *Methylobacter* A clade. Interestingly, we found the highest number of gene transcripts for the Methylococcales fermentation genes in the deepest sampled water depth where anoxic, methane-rich conditions prevailed, concomitant with the highest relative abundance of the respective MOB bins and transcription of *pmoABC* genes. These new data thus provide additional strong support for our conclusion about fermentation-based methanotrophy being employed by aerobic MOB in anoxic depths.

To show fermentation by Methylococcales, it would also be cool to show that the ¹³C-label of methane goes into organic acids (via compound specific isotope analyses, LC-IRSM) but that might be challenging as organic acids are expected to be readily consumed in mixed microbial communities.

We have measured samples on the LC-MS but were not able to detect ¹³C-methane-derived organic compounds in the incubation water. Indeed, it is very likely that the microbial community degrades them rapidly. Correspondingly, our nanoSIMS data showed uptake of methane-derived organic carbon by other, presumably non-methanotrophic bacteria in the incubations (see for example new figures S8 and S9).

Major Comment 2:

Whether or not Methylococcales need any oxygen to activate methane and drive methane oxidation is an interesting question. The methane monooxygenase enzyme is suggested to need oxygen to function.

In introduction and discussion, authors provide explanations on processes that could provide oxygen to methanotrophs living in anoxic water layers, like episodic oxygen intrusion from oxic layers (shallow and deep lakes) and photosynthesis (shallow lakes). The authors also acknowledge that they cannot completely rule out the possibility that some trace oxygen contamination could have affected their experiments.

I would like to know if authors would regard the recent findings on dark oxygen production relevant for their study: Some methanotrophs (alphaproteobacterial at least) produce methanobactins which when reacting with metals (like Fe³⁺) drive a water splitting reaction which produces free oxygen that can support methane oxidation: <https://journals.asm.org/doi/10.1128/aem.00286-21>

Could it sustain methane oxidation in the incubations of this study? By adding nitrate (which is readily reduced to nitrite in denitrifying conditions) one could actually enhance abiotic oxidation of Fe²⁺ to Fe³⁺ (coupled with nitrite reduction). The produced Fe³⁺ could then enhance the mentioned water splitting process. I actually wonder if the results by Oswald et al. 2016 on the addition of Fe oxides in enhancing methane oxidation in anoxic water samples of the study lake could be actually due to the water splitting process?

Furthermore, some ammonium oxidizing archaea can produce free oxygen from oxides of nitrogen in anoxic conditions: https://www.science.org/doi/10.1126/science.abe6733?url_ver=Z39.88-2003&rfr_id=ori:rid:crossref.org&rfr_dat=cr_pub 0pubmed

Could it sustain methane oxidation in the incubations of this study? By adding nitrate, one could as well enhance this process.

This is an interesting question. While we do not exclude these processes taking place in the anoxic waters of Lake Zug, we do not think that they represent a major source of oxygen, based on e.g. the low abundance of the respective organisms. We have added a sentence to the main text about these alternative mechanisms as potential biological sources of oxygen in anoxic waters (line 328ff), and we discuss these more extensively in the Supplementary Note 5.

Minor comments:

Lines 59-66. What about recent data indicating that methane oxidation by Methylococcales could be also coupled with reduction of iron and organic EAs. See e.g., <https://pubs.acs.org/doi/full/10.1021/acs.estlett.0c00436>

We can imagine that this could be potentially relevant for sediments or lakes with higher iron and humic acid contents. While we of course cannot exclude this proposed process in Lake Zug, we do not think that it is employed by the MOB in the anoxic waters, due to generally low abundance of

oxidized iron minerals in these anoxic depths (Fe concentrations were reported previously in Oswald et al., 2016). In any case, we now mention this mechanism in the revised manuscript.

Lines 86-89. I doubt if you can determine the methane consumption depths only using the concentration data? Would $\delta^{13}\text{C}$ of methane be a helpful addition to this? Furthermore, what is the accuracy of methane conc measurements? Were the concentrations shown in figure based on replicate measurements?

Methane concentration profiles can be used to determine zones of net methane consumption if there is no evidence of substantial lateral water transport or *in situ* production. Our data do not indicate such extensive mixing at these depths. We now clarify this in our respective statement (line 103).

Unfortunately, we do not have $\delta^{13}\text{C}$ methane measurements from this campaign. However, $\delta^{13}\text{C}$ measurements are less powerful at detecting methane oxidation that proceeds at low rates in waters with high background methane concentrations, such as in the deep anoxic hypolimnion (see Oswald et al., 2016). We would like to point out that our three incubation depths for methane oxidation measurement were chosen randomly as we only see oxygen profiles in real time while sampling.

The methane concentration measurements shown in figure 1a are based on single measurements from 12 discrete depths. The accuracy of the methane measurements using a gas chromatograph is 6% between duplicate samples. We now mention this in the methods (line 455).

Fig 1. Is the 1b the hypoxic treatment? It is described as “aerobic”, which is of course correct but for clarity it would be better mention also the word “hypoxic” here.

We thank the reviewer for pointing that out. We have now changed this to “microaerobic”, to be consistent with “anaerobic” in panel c.

Lines 102-103, 108-110 and other relevant parts. It would be good to actually see the accumulation data of $^{13}\text{CO}_2$ to confirm the finding that $^{13}\text{CO}_2$ production started after lag-phase in some treatment and without lag phase in other treatment. Can it be shown in supplementary figure?

We have added a time course to the Supplement, as suggested (new Figure S3).

Line 235-239. Indeed, strain *Methylobacter tundripaludum* SV96 is rod. But, the recent lake isolate by Khanongnuch et al. (2022), i.e., *Methylobacter* sp. S3L5C is actually cocci. And so is *Methylobacter psychrophilus* Z-0021, which is close relative to S3L5C. According to Khanongnuch et al. analyses, the strain S3L5C represents a large cluster of *Methylobacter* spp. present in lake water columns. Hence, *Methylobacter* in lake water columns can be also cocci. Can you be sure that the rods were all *Methylobacter*? Could there be additional analyses, like qPCR, to show that the cell number increase of rods is correlated with increase in *Methylobacter*?

We retrieved multiple *Methylobacter* 16S rRNA gene sequences as well metagenomic bins from our samples and the *Methylobacter*-related 16S rRNA gene sequences cluster closest to *Methylobacter tundripaludum*, which is a fat rod-shaped cell. However, we acknowledge that the shape alone would not be sufficient to unambiguously identify the large rods as *Methylobacter*. Therefore, we now performed additional analyses, as suggested. We performed new FISH analyses using a *Methylobacter*-specific probe (probe MLB482; Gullede et al. 2001); the same probe has been used by van Grinsven et al (<https://doi.org/10.1002/lno.11648>) to target *Methylobacter* cells. In their study, MOBs were rod-shaped, 2-3 μm in length and found in large and small clusters and thus similar to our large rod-shaped cells. These new analyses strongly support our original conclusion that the ‘large rods’ in our samples are representatives of the *Methylobacter* genus. Hence, we think it is valid to tentatively identify these cells as ‘Methylobacter-like’ MOB and we keep this in the manuscript.

We have additional unpublished data to support this claim. When we incubated water from Lake Zug (sample collected in October 2018 from 170 meters) under anoxic, nitrate-replete conditions for longer periods of time, we obtained a highly-enriched culture of large rod-shaped cells, which 16S rRNA gene was assigned to *Methylobacter* sp. The fact that they were readily enriched from lake water containing many different MOB confirms the ecological success of these *Methylobacter*-like cells under anoxia.

Fig. 1. Fluorescent microscopy images from an enrichment culture obtained from Lake Zug after circa three months of incubation under anoxic, methane and nitrate-replete conditions.

Line 268-271. Actually, I think that Kalyuzhnaya et al. 2013 suggest that the fermentation mode leads to decreased cellular assimilation and CO₂ production and increased excretion of fermentation products, like VFAs.

We have now deleted the word ‘efficiently’ and we also include a more detailed description of the proposed fermentation-based methanotrophy process (line 334).

line 407. Pure ¹³C-CH₄? Can you specify the ¹³C-label percentage? Not completely 100%?

We spiked our incubation bottles with 99 % purity ^{13}C -labeled methane. This information has been added to the Material and Methods. As incubation bottles were degassed prior to the experiments to remove oxygen, residual methane was removed as well, resulting in a ^{13}C labeling percentage of > 98%. We amended this in the methods section, see lines 497-500.

line 409. Did you measure the dissolved CH_4 conc in incubation bottles?

Yes, we measured the dissolved methane concentration in our incubation bottles at T0 (see lines 497).

line 422. Did you include non- ^{13}C – labeled controls to assess the background change in ^{13}C - CO_2 ? I assume that adding nitrate will anyway enhance production of CO_2 from organic matter oxidation. In that process, also natural ^{13}C - CO_2 is produced. How did you determine which part of the increase in ^{13}C - CO_2 is from your added ^{13}C - label and which part is increase in natural ^{13}C - CO_2 ?

No, we did not conduct control experiments without ^{13}C -methane. Oxidation of organic matter would produce both $^{12}\text{CO}_2$ and $^{13}\text{CO}_2$ at a fixed ratio. In contrast, in our incubations with $^{13}\text{CH}_4$ we see a large increase of $^{13}\text{CO}_2$ relative to $^{12}\text{CO}_2$, which can only result from ^{13}C -methane oxidation (only $^{13}\text{CH}_4$ was present in the incubations). In principle, fractionation can cause per mil changes in the $^{13}\text{C}/^{12}\text{C}$ ratio of the produced CO_2 but this would not be detectable on top of the per cent changes in $^{13}\text{C}/^{12}\text{C}$ caused by $^{13}\text{CH}_4$ oxidation.

Line 423-425. How did you make sure that no unwanted O_2 contaminated the incubations during these samplings?

Sample water was transferred in an anaerobic hood under N_2 / CO_2 atmosphere. Incubations were carried out with a headspace (30 ml helium headspace on top of 220 ml sample water) and were additionally degassed with helium for 15 minutes. For the sampling, we used a gas-tight glass syringe that was flushed with helium gas before insertion into the incubation vial. Still, we believe traces of oxygen (below the detection limit of our oxygen sensors) were present in the incubation bottles (see Supplementary Note 5 for a more in-depth explanation).

Line 436. Can you explain what is the notation “ $^{13}\text{CO}_2/^{12}\text{CO}_2 + ^{13}\text{CO}_2$ (in ppm)”?. Ratio of $^{13}\text{CO}_2$ to $^{12}\text{CO}_2$ plus concentration of $^{13}\text{CO}_2$?

We have now amended the calculation in the Material and Methods (lines 533-535).

Figure 4. Figure caption: “for 24 hours”. Unclear, since the data in figure goes until 5 days.

We have corrected this.

Reviewer #3 (Remarks to the Author):

Summary

The authors investigated the assimilation and oxidation of ^{13}C -methane of aerobic methanotrophs from Lake Zug under suboxic and anoxic+nitrate in short-term incubations over 5 days. They measured an oxygen profile depth and sample between 115 m and 180 m at 5-10 m intervals for chemical parameters methane, nitrate and nitrite. Incubation and CARD-FISH samples were collected from three depths, 123 m, 135 m and 160 m, DNA was collected from 125m, 135m and 160m respectively. In 2018 and 2019 incubations were done from deeper samples ($\geq 160\text{m}$ -180m, 175m-190m respectively). In 2018 and 2019 chemical profiles, bulk methane oxidation and bulk assimilation under oxic and anoxic+nitrate conditions were done, but not DNA, denitrification, CARD-FISH or nanoSIMS analysis. The authors targeted aerobic methane-oxidizing Gammaproteobacteria with CARD-FISH. The targeted methanotrophs were categorized into small or large rods, cocci and filaments, and cluster-forming coccoid cells. The incubations were done with ^{13}C -methane under suboxic ($10\ \mu\text{M O}_2$) or anoxic (no O_2 added, but 15N -nitrate added) conditions for a total of 6 incubations per year. The authors monitored the O_2 concentration in the incubations (data not shown). Five timepoints over 8 days were taken for bulk ^{13}C - CO_2 production and denitrification rates, as well as single cell analysis with nanoSIMS and bulk methane-C assimilation of filtered cells.

The authors found that large rods identified as Methylococcales by CARD-FISH continued to assimilate methane derived carbon under anoxic nitrate amended conditions at similar rates than under oxic conditions, which was also the case for some Crenothrix filaments (identified by morphology), but was not the case for cocci and small rods. The authors suggest that fermentation and denitrification may be used by these MOB, which are mechanisms that have been shown in previous studies to be oxygen saving mechanisms in aerobic methanotrophs, although for the initial methane oxidation oxygen was still needed in those experiments which also the authors acknowledge in line 65.

The authors reconstructed bins of Methylobacter, KS41, Methylovulum, UBA4132, UBA10906, SXIZ01 and Methylomirabilis an anaerobic methanotroph producing O_2 from nitrite which is then used to oxidize methane. All but KS41 seem to have nitrate reduction capability. Fermentation genes were not analyzed.

The authors convincingly show that large rods within Methylococcales continue to assimilate methane derived carbon under anoxic+nitrate conditions with very low oxygen concentrations. Therefore, either the large rods have substantial ability of saving oxygen by known (nitrate reduction, fermentation) or unknown mechanisms, produce their own oxygen (which is not known for Methylococcales), or use fully or partially other electron acceptors (nitrate reduction or else), or other unknown mechanisms – what the mechanisms are remains a question to be answered.

Major comments

1) *Methylomirabilis*

In line 160 they mention that *Methylomirabilis* was found in the DNA data (a bin was recovered), which is not surprising given that a subset of the authors has found abundant *Methylomirabilis limnetica* (up to 27%) in an earlier study of the same lake. This does complicate the interpretation of the following parameters: a) bulk ^{13}C - CO_2 production which could come fully or partially from *Methylomirabilis* in the sample. b) Similarly, bulk ^{13}C -methane assimilation might contain the anaerobic methane oxidizer *Methylomirabilis*. This seems likely as the anoxic incubations were amended with nitrate. c) $^{15}\text{N}_2$ production which is also possible to be formed by *Methylomirabilis*. It is somewhat surprising that the presence of *Methylomirabilis* is not discussed and shown throughout the manuscript.

I strongly suggest to include the potential role of *Methylomirabilis* throughout the manuscript which might change several conclusions for bulk methane oxidation rates and assimilation rates. Further *Methylomirabilis* should be included in the figures of the DNA results (e.g. Fig. 2a,bc and Fig. 5) and a abundance depth profile should be included analogous to aerobic MOB in Fig. 2c to be able to assess the abundance of this anaerobic methanotroph, which is currently not possible. In case *Methylomirabilis* has much lower abundance than *Methylococcales* still a potential contribution should be discussed.

We have now performed new analyses to consider the role of *Methylomirabilis* bacteria in Lake Zug during our sampling campaign (September 2017). In contrast to the exceptional situation described in Graf et al., 2018, the relative abundances of *Methylomirabilis* bacteria in the hypolimnion was low (less than 1.5%; based on 16S rRNA data) (Data file S5). The absolute cell counts were in the range of ca. 4.6×10^2 cells ml^{-1} at 123 m and increased to ca. 6.0×10^3 cells ml^{-1} at 160 m (Fig. 1e), which is about 4-times less than the gamma-MOB abundance but similar to the large rods. The average biovolume of these methanotrophs is with $0.14 \mu\text{m}^3$ substantially lower compared to $7.4 \mu\text{m}^3$ of the large rods.

We analyzed *Methylomirabilis* cells from our anoxic, ^{15}N -nitrate-replete incubations with nanoSIMS and based on their ^{15}N enrichment and low ^{13}C enrichment we conclude that these cells were active in our incubations but – like suggested previously – do not grow on methane as preferred carbon source (Rasigraf et al. 2014; doi: <https://doi.org/10.1128/AEM.04199-13>). Based on these data we are confident that *Methylomirabilis* did not substantially contribute to the measured C assimilation rates and as such do not substantially contribute to MOB-mediated methane carbon retention in the lake hypolimnion, which is one of our main conclusions. Similarly, their low abundance and small biovolume compared to the four investigated gamma-MOB groups indicates only a minor

contribution to the measured methane oxidation and denitrification rates during our 2017 campaign. We have now included all these data into the figures (Figure 1, 2, 5, 6 and Supplementary Fig. S7) as well as a more thorough discussion of *Methylomirabilis* and its role in Lake Zug at the time of sampling (e.g. lines 309-315, lines 343-349).

Growth rates based on morphology and CARD-FISH

According to S3 the large rods did not increase their cell numbers over time, but rather stayed the same from day1 to day2 and then declined slightly to day5. See methods to clarify how the growth rates were calculated and potential issues.

We now report cell counts from day 8 of the incubations, which show increase in cell numbers at this time point. We would like to point out that with the estimated growth rates (0.3 to 0.4 based on nanoSIMS data), the increase in cell numbers for the first time points would be within the margin of error of counting.

2) Oxygen contamination

The authors monitored the oxygen in the incubations over time, which is not usually done but is very interesting important information, but currently the data is not shown. I suggest to show the data in the manuscript because it can rule out presence of substantial oxygen concentration, which often is difficult to avoid and e.g. taking timepoints, or oxygen contaminated gases ($^{13}\text{C-CH}_4$, He) could introduce oxygen. The purity of the used gases, manufacturer and LOT nr (He, $^{13}\text{C-CH}_4$) should be stated including the contamination levels of oxygen if available, and how $^{13}\text{C-CH}_4$ was taken from the bottle without introducing oxygen. This information will be of interest for other researcher who would also like to study aerobic methanotrophs under oxygen starvation.

We want to clarify that we measured oxygen concentrations in our incubation bottles at each time point, with discrete measurements with the optode reader. These read-offs were done prior to and after sampling to assess consumption of oxygen (in microaerobic incubations) or potential oxygen contamination (in anaerobic incubations). These were not continuous measurements and as such we did not monitor oxygen concentration over time. The oxygen concentrations from these discrete measurements from the microaerobic incubations are included in the Material and Methods. We have now added a sentence clarifying that oxygen concentrations in the anoxic incubations were below the detection limit of the oxygen sensor at all sampling time points (line 513).

As suggested by the reviewer, we have added the requested information on the helium and $^{13}\text{C-}$ methane gas bottles. We have included a description of how $^{13}\text{CH}_4$ was sampled from the bottle (lines 500ff).

3) Morphological characterization and nanoSIMS result

The authors show one example of each category in Fig. 3a, b. The main shape “large rod” is only shown once and in my opinion, it would be important to show more than one image of such a cell. It is crucial to also show the CARD-FISH + nanoSIMS image of the anoxic+nitrate active cell and not just the oxygen+methane incubation images.

As suggested by the reviewer, we have now included more CARD FISH images and corresponding nanoSIMS measurements of all MOB types from anoxic incubations. Examples of large rod-shaped cells are included with the main Figure 3 and 4, other examples of also the small rods, filaments, and cocci in anoxic incubations are now included as supplementary figures S8 and S9.

4) DNA results/Phylogenetic analysis

The title of the manuscript states “Methylobacter-like” aerobic methanotrophic bacteria for the “large rod shape” Methylococcales. I think the taxonomic identification “Methylobacter-like” is not justified based on the data presented. a) multiple taxonomic groups within Methylococcales are present including several uncultivated genera (6 genera Methylobacter_A (3bins) SXIZ01(1bin) KS41 (3bins), Methylovulum (1bin), UBA4132 (2bins) and UBA10906 (3 bins)) making it impossible to identify which one is the “large rod”. In my opinion shape is not a good indicator in here especially since the shape for several uncultivated groups is unknown b) for identifying the organism further analysis e.g. a targeted FISH probe would be needed. Therefore, I suggest removing the conclusion Methylobacter throughout the manuscript.

We removed ‘*Methylobacter*-like’ from the title of our paper because we do not want the uncertainty regarding the name of the large rod-shaped MOB to distract from the main messages of our manuscript. We are aware that the name *Methylobacter* might encompass different MOB as a result of the fragile MOB phylogeny and the polyphyletic nature of this group (see also answer below). Still, ‘*Methylobacter*’ MOB are currently commonly referred to in the literature and we would like our study to relate to these studies. Therefore, we still tentatively identify these cells as ‘*Methylobacter*-like’ MOB in the main text. This phylogenetic identification is supported by multiple lines of evidence:

1. The presence of *Methylobacter* MOB could be confirmed with both Methylococcales genome and 16S rRNA phylogeny.
2. Abundance of *Methylobacter*-related bins increased below the oxic-anoxic interface, as did the number of large rod-shaped MOB.

3. *Methylobacter*-related 16S rRNA gene sequences cluster closest to *Methylobacter tundripaludum*, which is a fat rod-shaped cell.

4. We performed new FISH analyses using a *Methylobacter*-specific probe (probe MLB482; Gullede et al. 2001). The same probe has been used by van Grinsven et al. 2020 (<https://doi.org/10.1002/lno.11648>) to target *Methylobacter* cells. In their study, MOB were rod-shaped, 2–3 μm in length and found in large and small clusters and thus similar to our large rod-shaped cells. These new analyses strongly support our original conclusion that the ‘large rods’ in our samples are representatives of the *Methylobacter* genus.

5. We have additional unpublished data to support this claim. When we incubated water from Lake Zug (collected from 170 m depth in October 2018) under anoxic, nitrate-replete conditions for longer periods of time, we obtained a highly-enriched culture of large rod-shaped cells, which based on 16S rRNA gene phylogeny belonged to the genus *Methylobacter* (see figure 1 included with our previous answer to reviewer 2).

Figure 2: The phylogenetic analysis of the aerobic methanotrophs present is very important, since the claim is that one “group” but not all aerobic methanotrophs are capable of some continued anaerobic activity. The 16S rRNA tree in Fig. S2 does not suggest that e.g. GammaMOB1 160m or 123m is *Methylobacter* since the clades are not significantly. The lacustrine *Crenothrix* is supported and the CABC2E06 is supported too. There are sequences in 3 additional clades without a taxonomic identification.

In Figure 2b 6 methanotrophic groups are shown. *Methylobacter* is not a valid aggregation based on tree S2. *Methylomarinum* in that figure does not occur in the tree S2, neither does *Methyloglobulus* which is surprising. Further, it is unclear which sequences are included in “uncultured” vs “other *Methylococcales*”. I suggest using the classification of the tree instead, supported are “*Crenothrix*” and “CABC2E06”. E.g. by looking at significance lower than 90% like 80 or 70% in the tree together with % identity between sequences a better classification can be found (general classifiers sometimes perform poorly on genus level and a tree should be made). Since it is unclear if the “anoxic” trait is at strain/species/genus level, and there are only about 10 16S rRNA sequences showing the depth distribution of all 10 sequence variants instead of the grouping in Fig. 2b) would be very useful. In case the bins have 16S sequences it could be tried to connect the classifications.

We agree with the reviewer that the different classification in the 16S rRNA and whole genome tree was confusing and we therefore re-ran the 16S tree including more reference sequences and following the same classification using the GTDB database. We included more reference sequences into the 16S rRNA gene tree to better resolve the phylogenetic relationship of the 16S sequences in our samples. As such, the respective taxonomic groups are now directly comparable between the

16S rRNA gene (included as data file S2) and whole genome tree (Figure S5). The comparison shows that the closest genera in both trees are consistently the same ones, including *Methylobacter* A, KS41, and the UBA clades. It should be noted that the *Methylobacter* group appears polyphyletic in both trees, in agreement with previous reports (<https://doi.org/10.3389/fmars.2017.00023>).

Regarding the depth distribution of these 16S rRNA gene sequences, we want to note that each sequence was only retrieved from a single depth, meaning we cannot show their depth distribution. However, we have now included the respective depth for each of these ten 16S sequences in the tree. The bins, unfortunately, did not contain 16S sequences.

Minor comments

Titel and Abstract, please see major points.

We have now removed the reference to *Methylobacter* from the title, as suggested.

Introduction

Line 30-32 substantiate with citations.

We have now added references for these statements.

39 consider mentioning the proposed mechanism of *Methylomirabilis*

This has been amended (see lines 45-48).

69 e.g. *Methylomirabilis* is an anaerobic bacterial methanotroph, consider formulating more specific

This has been changed accordingly.

Results

Figure 1 b, c, please consider using same scale for aerobic and anaerobic MO for a better comparison.

We realize that the same scales would make visual comparison easier but one would not be able to read off the values for anaerobic oxidation rates from the graph. As these rates are central to this manuscript, we chose to leave the scale bars unchanged. The difference between the scales is pointed out in the figure legend.

Figure 1 e, showing small rods as x10 is unusual and makes it difficult to read. Consider showing all with same scaling or show the biovolume data, or two figures as a suggestion.

We have now revised this figure.

Figure 2 needs major revision see major points

This figure has been revised.

Figure 3 please show all data points, the asterisk is not acceptable. I am surprised the authors consider *Crenothrix* under anoxic+nitrate condition an outlier since Oswald et al 2017 has shown a similar result for *Crenothrix* from Lake Zug.

We apologize for the confusion. The high *Crenothrix* ^{13}C values indicated with an asterisk were included in our data analyses, they were only omitted from the plot in Figure 3c to improve the readability of the graph. Nevertheless, we have now added them to the figure (now Figure 2b).

114 discuss further in discussion section, please clarify why a correlation was expected? It seems aerobic methanotrophs do not possess *nosZ* usually. Were the authors trying to measure *Methylomirabilis* activity 30N_2 ? Please consider discussing also non methanotrophs as N_2 producers.

The reviewer is right and we recognize that this argument needs a clarification, which we now included in the revised manuscript. Briefly, we assumed rapid turnover of the produced N_2O to N_2 by N_2O -reducing (non-methanotrophic) bacteria. However, as the reviewer correctly points out, also denitrification by heterotrophic microorganisms and *Methylomirabilis* spp. would be detected and we now discuss this in our manuscript (line 386-391).

115 The N_2O production is interesting please consider showing the data.

As we originally planned to only measure $^{15}\text{N}_2$ production (data shown in Figure 1) we unfortunately did not add a $^{14}\text{N}_2\text{O}$ pool in our incubations to trap the produced $^{15}\text{N}_2\text{O}$ before it gets reduced to $^{15}\text{N}_2$ (see response to comment above). Without this, we cannot properly quantify N_2O production rates over time, even though we can confidently say that $^{15}\text{N}_2\text{O}$ was produced in the incubations.

122 Please compare to literature values. Please consider that production of dissolved organic carbon is not included here (e.g. formate, acetate, methanol, formaldehyde, lactate) which might play a role.

We have now included a sentence stating that some methane carbon is also converted into DOC that is excreted (line 344).

137 it seems the increase in abundance with depth here is not as pronounced as in the DNA data. Consider to discuss.

The data in (new) figure 5a corresponds to relative abundances (determined from molecular data) and the values presented here refer to absolute abundances (determined from FISH analyses) and are not directly comparable. DNA-based estimates of abundances are known to be biased due to e.g. bacterial polyploidy, DNA extraction biases, multiple 16S gene copy numbers etc. We now mention

in the text that e.g. multiple 16S rRNA operons in some bacterial species may lead to an overestimation of their relative abundance using molecular methods (line 216).

142 Because *Crenothrix* was not identified with a probe either separately or in the mix, it cannot be entirely excluded that the low activity cells under anoxic conditions may not be *Crenothrix*. Please add to discussion.

This is true and we have now added this point in the discussion (line 300).

156 see major comments on *Methylobacter*

Please see answer to previous comment.

167 since fermentation is a big discussion point, consider analyzing potential pathways

As suggested, we have now performed new metagenomic and metatranscriptomic analyses and included an overview of the fermentation genes detected in the different bins along with their expression in the revised manuscript.

184 did you consider sequencing the bottle, it might give an indication which methanotroph was growing.

Unfortunately, this was not possible due to the small volumes of the incubations at the end of the experiment (ca. 60 ml). However, as mentioned in our answer to major comment about *Methylobacter*, in the past we obtained a highly enriched culture from the anoxic lake waters that consisted mainly of large rod-shaped cells, which affiliated with *Methylobacter* based on the 16S rRNA gene analysis.

186 according to Table S3 the cell counts of large rods did not increase, since no standard deviations are given I would expect to see an increasing trend from day1 to day5 which seems not to be the case. Especially because this is one of the main results. Please include day 0 in Tables S3 so a comparison can be made.

We have now included cell counts from day 8 that show an increase in large rod cell abundances as compared to the start of the experiment (T0). We would like to point out that with the estimated growth rates (0.3 to 0.4 based on ¹³C data), the increase in cell numbers for the first time points would be within the margin of error of counting.

190 did you see the large rods after 8 days as well?

Yes, we did. These data are now included (line 169).

200 consider moving this to discussion, since no oxygen reductant was added residual oxygen is an explanation too

We have now included here a reference to Supplementary Note 5, where we discuss the role in of residual oxygen in the incubations.

Discussion

Please consider incorporating more examples of aerobic methanotrophs in anoxic habitats. This is observed quite often and has been studied in some other lakes and other environments e.g. marine too. E.g. Su, G., Lehmann, M.F., Tischer, J. et al. Water column dynamics control nitrite-dependent anaerobic methane oxidation by Candidatus “Methylomirabilis” in stratified lake basins. ISME J 17, 693–702 (2023). <https://doi.org/10.1038/s41396-023-01382-4>

We have now included more recent references.

220 the bathymetry of a lake can also influence the methane profile.

We have now reformulated this sentence.

228 Figure 1e only shows aerobic methanotrophs, Methylomirabilis or Methanoperedens are not shown in the Figure. Please add Methylomirabilis and check if other archaeal methanotrophs are present.

As suggested, we have now added *Methylomirabilis* cell counts to Figure 1e. We did not detect archaeal methanotrophs in our molecular data (only a few read assigned to ANME archaea; see new data file 5), indicating that they are essentially absent from the water column.

245 please clarify somewhere why nitrate was added to anoxic incubations

We have included this information in the text (line 115, line 541).

236 see major point “Methylobacter”

Please see answer to major comment *Methylobacter*.

252 please cite and compare to finding of Oswald, K., Graf, J., Littmann, S. et al. Crenothrix are major methane consumers in stratified lakes. ISME J 11, 2124–2140 (2017). <https://doi.org/10.1038/ismej.2017.77>, who also showed activity of Crenothrix under anoxic+nitrate conditions in lake Zug.

We now discuss the activity of *Crenothrix* in more detail in Supplementary Note 3.

Note 3: 13 bins are shown in Fig.2, in Note 3 14 are mentioned, please clarify

This is because bin 14 was assigned to *Methylomirabilis* and this figure originally only showed the gamma-MOB bins. However, all 14 bins are now shown in Figure 5b.

255 the authors do not quantify *Methylomirabilis* here

As suggested by the reviewer, we have now added FISH and nanoSIMS measurements of *Methylomirabilis* and the *Methylomirabilis* bin properties into the revised manuscript (see figures 1, 2, and 5).

276 fermentation by methanotrophs may occur, please phrase more carefully. Fermentation by methanotrophs in Lake Zug was not shown. The mechanism in my opinion is unknown, it could be trace amount of oxygen, nitrate reduction or fermentation or an as yet unknown mechanism.

We have now added additional molecular data strengthening our argument for the role of fermentation-driven methanotrophy in anoxic depths, but we agree with the reviewer that likely a multitude of mechanisms is employed by these aerobic bacteria under anoxic conditions and we have reformulated this paragraph accordingly.

278 volatile fatty acids have not been mentioned before in the manuscript, please include in introduction and explain in detail including references or consider leaving it out.

We have now included a sentence in the introduction clarifying that fermentation-based methanotrophy would lead to the production of volatile fatty acids (see line 74).

283 Consider revising the conclusion that fermentation has great relevance. The experiments done in this study do not allow such a conclusion, but the authors incubated with additional nitrate and showed N₂ and N₂O production which are interesting results and N₂O production could be discussed instead.

We have now added new molecular data, which strengthen our conclusion about the importance of fermentation-driven methanotrophy. At the same time we have also added an additional paragraph discussing other potential mechanisms, including denitrification. Regarding denitrification, more information was added about N₂O production in our incubations and the N₂O production rates have been included as new Supplementary Fig. S4.

286ff Please mention that this is also the case under oxic conditions.

We have now included this (line 363).

297 please consider also discussing N₂O production, since N₂ production is not expected from *Methylococcales*

We have included more information on N₂O production and potential contribution from MOB (lines 386ff).

Methods

364 consider showing a conductivity profile since it is interesting to see the stratification.

Unfortunately, we don't have a conductivity profile from this campaign. However, we do not expect a strong salinity gradient in this freshwater lake.

393 are cluster forming coccoid cells and cocci the same? I only see cocci in all figures, consider adding the dimensions of the cells also here.

We have now clarified in the methods, that cocci encompass any free-living coccoid cell as well as cocci arranged in clusters (line 482). The size and biovolume of all cell types are included in supplementary table S2.

401 please add how much volume was added to reach 20 uM nitrate and please clarify why nitrate was added in the text.

This information has now been included into the methods (line 519).

402 please add manufacturer and purity for all gases including the N₂/CO₂ mix, He and 13C-methane. What kind of O₂ scrubber was used? Please clarify in text.

We have now added information regarding the gas purity. The anaerobic chamber contains a copper catalyst for oxygen stripping and oxygen is continuously monitored in the chamber.

409 the concentration in water seems too low considering 5ml CH₄ in a 30ml headspace. Please clarify. Where does the variability originate from? Please clarify in text.

The measured concentrations are lower than what was expected; however, this is very likely due to the losses of gas during transfer from the container to incubation bottle. Also, it is possible that the methane has not fully equilibrated at the time of sampling. We have now included this into the methods (line 505).

411 in the supplements 20 uM are stated. Where does the variability come from? Please clarify in text.

We apologize for the mistake, the oxygen concentrations were circa 10 μM, this is now consistent throughout the manuscript and the supplementary files.

416 oxygen was monitored see major point consider including the data

Please, see answer to previous comment.

472 please clarify how many cells were counted to determine the numbers in table S3.

This information has been added (line 482).

514 In my opinion using the start and end point here is only valid if there was exponential growth observed and expected, I do not think this kind of calculation is valid for the anoxic incubations, since Table S3 shows that there was no exponential growth but rather a stable to decline of cell numbers for all categories including the large rods. To me is unclear how a positive growth rate was calculated? Please provide the cell numbers from the beginning of the experiment in table S3. Even if t_0 was lower, it looks like the large rods grew less and declined a bit towards day 5 and the exponential equation is probably not a good fit.

As far as I understand the incubation water was also stored for a week until incubation, and the I assume the Card-FISH cells were fixed and filtered immediately after sampling? Please clarify in the methods which cell number was used for t_0 in the calculation. If the value from immediately fixed cells was used, there is a chance the large rods grew in that period. So I think using day 1 and day 5 or 8 if available would be more suitable to evaluate what was growing under the respective conditions.

If the large rods did not increase their cell number, there is the question why the assimilation rates were so high nevertheless. I think that is something that should be discussed.

As suggested, we have now included cell counts from day 8 which show increase in cell numbers of large rods under anoxic conditions and we have now added more information regarding the sampling for FISH and cell counting. We now also discuss the most likely reasons for the delayed increase in cell numbers over time despite immediate uptake of ^{13}C . Briefly, the nanoSIMS has a very high sensitivity of ^{13}C detection and therefore growth can be detected even before it manifests as doubling of cell numbers, which can be masked by simple error of counting. Additionally, grazing and lysis are bound to negatively affect cell counting as a proxy for growth. These considerations are included in the revised manuscript (Supplementary Note 4).

576 consider attaching the phyloflash result so an overview of the microbial community can be gained.

We have now uploaded this data as an additional file (Data file S5).

Reviewer #1 (Remarks to the Author):

Dear authors, Dear editor,

I am fully satisfied with the revision of the manuscript.

The authors did an excellent job, providing a great deal of new metagenomics and metatranscriptomic data as well as single cell imaging in support of their claims of fermentation based methanotrophy in Lake Zug. The addition of new text and explanations clarified various aspects and concerns raised by this reviewer.

I have no further concerns and thus I recommend the manuscript for publication.

Kind regards
Niculina Musat

Reviewer #2 (Remarks to the Author):

I acted as the reviewer #2 during the first round. In this revision, authors have done a great job in addressing my comments. I have only few minor points. Great work!

Lines 78-79. About highlighting the broad prevalence of fermentation-based metabolism of MOB based on metagenomics. Although it can be quite suspicious when a reviewer recommends citations to his own papers, in this case I truly objectively think that further citations to two of our other papers would be even better here than the current one (which is also ours). In the suggested papers, the spatial coverage of metagenomic analyses showing fermentation genes in genomes of lake and pond MOB is larger (the results are shown in supplementary files in the papers). Hence, if you wish, you could consider citing also these:

<https://www.nature.com/articles/s43705-022-00172-x#Sec11> (this is already cited in other context)

<https://journals.asm.org/doi/10.1128/spectrum.01742-23>

In those papers, we also experimentally show that lake MOB isolates convert methane into VFAs in laboratory conditions. We have actually shown that also for *Methylobacter tundripaludum* SV96:

<https://www.frontiersin.org/journals/microbiology/articles/10.3389/fmicb.2022.874627/full>

but I think that it is not worthwhile to discuss our laboratory experiments in your paper since it is challenging to link observations in optimum laboratory conditions (our experiments) to those in situ (your work).

Line 330. If I am correct, Zheng et al. (citation 40) do not suggest that iron reduction would produce O₂. But, iron reacting with methanobactins could lead to water splitting and O₂ production as shown in the cited reference 55. Please, check this and delete citation if I am correct.

Line 441. Filters for RNA extraction stored at -20°C. It might have caused some loss in mRNA if they were not stored at -80 or with some RNA preserving preservative. But, in this case this is ok as clear results were obtained. Great!

All the best,
Antti J Rissanen (Academy Research Fellow, PhD, Tampere University, Finland)

Reviewer #3 (Remarks to the Author):

The authors thoroughly revised the manuscript. They added an additional analysis using metatranscriptomics supporting both fermentation and nitrate reduction as two plausible oxygen

saving strategies by gammaproteobacterial methanotrophs in Lake Zug, which makes the conclusions stronger. The authors also added the result from *Methylomirabilis* which gives a more comprehensive view of methanotrophs in that lake. I have a few comments on the revised manuscript below:

Additional comments:

Discussion

The metatranscriptomic analysis did not reveal the identity of the specialized subset of methanotrophs suggested to be active under anoxic conditions by nanoSIMS. Seemingly, several bins/genera were found to have and express the genes for both nitrate reduction and fermentation. Consider adding a sentence why that might be the case or what could be done in future to find out more about it. E.g. cultivation or enrichment – since the authors seem to have further results on that. Potentially also suggest some other analyses that would shine more light on methanotroph physiology under close to anoxic conditions.

Abstract

The authors should clarify in the abstract that oxygen is still thought to be required for methane oxidation by gammaproteobacterial methanotrophs. It might lead to misunderstandings otherwise. Line 22 should be clarified since methane oxidation was quite different between hypoxic and anoxic incubations and was not identical. Please clarify that the large rods showed similar methane assimilation based on nanoSIMS.

Response to reviewers

Reviewer #1 (Remarks to the Author):

Dear authors, Dear editor,

I am fully satisfied with the revision of the manuscript.

The authors did an excellent job, providing a great deal of new metagenomics and metatranscriptomic data as well as single cell imaging in support of their claims of fermentation based methanotrophy in Lake Zug. The addition of new text and explanations clarified various aspects and concerns raised by this reviewer.

I have no further concerns and thus I recommend the manuscript for publication.

Kind regards

Niculina Musat

Thank you for the positive feedback and your previous comments that helped us improve the manuscript.

Reviewer #2 (Remarks to the Author):

I acted as the reviewer #2 during the first round. In this revision, authors have done a great job in addressing my comments. I have only few minor points. Great work!

Lines 78-79. About highlighting the broad prevalence of fermentation-based metabolism of MOB based on metagenomics. Although it can be quite suspicious when a reviewer recommends citations to his own papers, in this case I truly objectively think that further citations to two of our other papers would be even better here than the current one (which is also ours). In the suggested papers, the spatial coverage of metagenomic analyses showing fermentation genes in genomes of lake and pond MOB is larger (the results are shown in supplementary files in the papers). Hence, if you wish, you could consider citing also these:

<https://www.nature.com/articles/s43705-022-00172-x#Sec11> (this is already cited in other context)

<https://journals.asm.org/doi/10.1128/spectrum.01742-23>

In those papers, we also experimentally show that lake MOB isolates convert methane into VFAs in laboratory conditions. We have actually shown that also for *Methylobacter tundripaludum* SV96:

<https://www.frontiersin.org/journals/microbiology/articles/10.3389/fmicb.2022.874627/full>

but I think that it is not worthwhile to discuss our laboratory experiments in your paper since it is challenging to link observations in optimum laboratory conditions (our experiments) to those in situ (your work).

Thank you for pointing this out. We agree with the reviewer that the suggested references are relevant for our discussion and included them into the revised manuscript (line 81).

Line 330. If I am correct, Zheng et al. (citation 40) do not suggest that iron reduction would produce O₂. But, iron reacting with methanobactins could lead to water splitting and O₂ production as shown in the cited reference 55. Please, check this and delete citation if I am correct.

We have checked the respective references and corrected the text accordingly.

Line 441. Filters for RNA extraction stored at -20°C. It might have caused some loss in mRNA if they were not stored at -80 or with some RNA preserving preservative. But, in this case this is ok as clear results were obtained. Great!

We agree that RNA samples should be stored at - 80°C to avoid loss of RNA yield. In our case, the samples were extracted within two months after sample collection. This information has been added to the revised Material and Methods section (line 452).

All the best,

Antti J Rissanen (Academy Research Fellow, PhD, Tampere University, Finland)

Reviewer #3 (Remarks to the Author):

The authors thoroughly revised the manuscript. They added an additional analysis using metatranscriptomics supporting both fermentation and nitrate reduction as two plausible oxygen saving strategies by gammaproteobacterial methanotrophs in Lake Zug, which makes the conclusions stronger. The authors also added the result from *Methylomirabilis* which gives a more comprehensive view of methanotrophs in that lake. I have a few comments on the revised manuscript below:

Additional comments:

Discussion

The metatranscriptomic analysis did not reveal the identity of the specialized subset of methanotrophs suggested to be active under anoxic conditions by nanoSIMS. Seemingly, several bins/genera were found to have and express the genes for both nitrate reduction and fermentation. Consider adding a sentence why that might be the case or what could be done in future to find out more about it. E.g. cultivation or enrichment – since the authors seem to have further results on that. Potentially also suggest some other analyses that would shine more light on methanotroph physiology under close to anoxic conditions.

As suggested, we added a sentence to the discussion.

Line 386ff: Interestingly, even though many gamma-MOB possessed and expressed genes related to denitrification (as well as fermentation-based methanotrophy), methane-dependent growth under apparent anoxia could only be observed for one discrete MOB morphotype (i.e. large rods). More cultivation-based as well as cultivation-independent studies are still needed to better understand the prevalence of anaerobic metabolisms of this environmentally relevant group of microorganisms.

Abstract

The authors should clarify in the abstract that oxygen is still thought to be required for methane oxidation by gammaproteobacterial methanotrophs. It might lead to misunderstandings otherwise.

We have added a sentence to the abstract to clarify this.

Line 22 should be clarified since methane oxidation was quite different between hypoxic and anoxic incubations and was not identical. Please clarify that the large rods showed similar methane assimilation based on nanoSIMS.

We have now clarified this in the revised manuscript.